# Bayesian Strategic Classification

Lee Cohen[1]    Saeed Sharifi-Malvajerdi[2]    Kevin Stangl[2]    Ali Vakilian[2]    Juba Ziani[3]

## Abstract

In strategic classification, agents modify their features, at a cost, to obtain a positive classification outcome from the learner's classifier, typically assuming agents have full knowledge of the deployed classifier. In contrast, we consider a Bayesian setting where agents have a common distributional prior on the classifier being used and agents manipulate their features to maximize their expected utility according to this prior. The learner can reveal truthful, yet not necessarily complete, information about the classifier to the agents, aiming to release just enough information to shape the agents' behavior and thus maximize accuracy. We show that partial information release can counter-intuitively benefit the learner's accuracy, allowing qualified agents to pass the classifier while preventing unqualified agents from doing so. Despite the intractability of computing the best response of an agent in the general case, we provide oracle-efficient algorithms for scenarios where the learner's hypothesis class consists of low-dimensional linear classifiers or when the agents' cost function satisfies a sub-modularity condition. Additionally, we address the learner's optimization problem, offering both positive and negative results on determining the optimal information release to maximize expected accuracy, particularly in settings where an agent's qualification can be represented by a real-valued number.

## 1   Introduction

Machine Learning critically relies on the assumption that the training data is representative of the unseen instances a learner faces at test time. Yet, in many real-life situations, this assumption fails when individuals (agents) manipulate decision-making algorithms for personal advantage, by modifying their features at a cost. A typical example of such manipulations or *strategic behavior* is seen in loan applications or credit scoring: for example, an individual may open new credit card accounts to lower their credit utilization and increase their credit score artificially. In the context of job interviews, a candidate can spend time and effort to memorize solutions to common interview questions and potentially look more qualified than they are at the time of an interview. A student might cram to pass an exam this way without actually understanding or improving their knowledge of the subject.

The prevalence of such behaviors has led to the rise of an area of research known as *strategic classification*. Strategic classification, introduced by Hardt et al. [2016], aims to understand how a learner can optimally modify decision making algorithms to be robust to such strategic manipulations of agents, if and when possible.

Most of the strategic classification literature makes the assumption that the model deployed by the learner is *fully observable* by the agents, granting them the ability to optimally best respond to the learner using resources such as effort, time, and money. Yet, this *full information* assumption can be unrealistic in practice. There are several reasons for this: some machine learning models are

---

[1]Stanford University. Email: leeco@stanford.edu

[2]Toyota Technological Institute at Chicago (TTIC). Email: saeed@ttic.edu, kevin@ttic.edu, vakilian@ttic.edu

[3]Georgia Institute of Technology. Email: jziani3@gatech.edu

38th Conference on Neural Information Processing Systems (NeurIPS 2024).

proprietary and hide the details of the model to avoid leaking "trade secrets": e.g., this is the case for the credit scoring algorithms used by FICO, Experian, and Equifax.[1] Some classifiers are simply too complex in the first place to be understood and interpreted completely by a human being with limited computational power, such as deep learning models. Other classifiers and models may be obfuscated for data privacy reasons, which are becoming an increasingly major concern with new European consumer protection laws such as GDPR [Regulation, 2018] and with the October 2023 Executive Order on responsible AI [Biden, 2023]. In turn, there is a need to study strategic classification when agents only have *partial* knowledge of the learner's model.

There has been a relatively short line of work trying to understand the impact of incomplete information on strategic classification. Jagadeesan et al. [2021] and Bechavod et al. [2021] study the optimal classifiers in settings where agents can only gain partial or noisy information about the deployed model. Haghtalab et al. [2023] study calibrated Stackelberg games, a more general form of strategic classification; in their framework, the learner engages in repeated interactions with agents who base their actions on calibrated forecasts about the learner's classifier. They characterize the optimal utility of the learner for such games under some regularity assumptions. While we also model agents with prior knowledge of the learner's actions, Haghtalab et al. [2023] focus on an online learning setting and the selection of a strategy for the learner without incorporating any form of voluntary *information release* by the learner.

In contrast, we focus on this additional critical aspect of voluntary *information release* by the learner that these works do not study. Namely, we ask:

**How to release partial and truthful information about the classifier to maximize accuracy?**

This should give the reader pause: why should a learner release information about their deployed classifier since presumably such information only makes it easier for agents to manipulate their features and "trick" the learner. In fact, Ghalme et al. [2021] showed that information revelation can help—a learner may prefer to fully reveal their classifier as opposed to hiding it. While they consider either fully revealing the classifier or completely hiding it, our model considers a wider spectrum of information revelation that includes both "full-information-release" and "no-information-release". We show that there exist instances where it is optimal to reveal only *partial* information about the classifier, in a model where a learner is *allowed to reveal a subset of the classifiers containing the true deployed classifier.* For example, a tech firm might reveal to candidates that they will ask them about a new type of data structure during their job interviews. Lenders might reveal to clients that they do not consider factors like credit score [Lake, 2024]. This selective disclosure can discourage unfit individuals, ultimately saving time and energy for both sides. In the following, we summarize our contributions.

**Summary of contributions:**

- In Section 2, we propose a new model of interactions between strategic agents and a learner, under partial information. The two novel modeling elements compared to the standard strategic classification literature are: i) agents have partial knowledge about the learner's classifier in the form of a *distributional prior* over the hypothesis class, and ii) the learner can *release partial information* about their deployed classifier.

  Specifically, our model allows the learner to release a *subset* of the hypothesis class to narrow down the agents' priors. Given our model, we consider a (Stackelberg) game between agents with partial knowledge and a learner that can release partial information about its deployed model. On the one hand, the agents aim to manipulate their features, at a cost, to increase their likelihood of receiving a positive classification outcome. On the other hand, the learner can release partial information to maximize the expected accuracy of its model, after agent manipulations.

- In Section 3, we study the agent's best response in our game. We show that while in general, it is intractable to compute the best response of the agents in our model, there exist oracle-efficient algorithms[2] that can *exactly* solve the best response when the hypothesis class is the class of low-dimensional *linear* classifiers. We then move away from the linearity assumption and consider

---

[1]"The exact algorithm used to condense your credit report into a FICO score is a closely guarded secret, but we have a general layout of how your credit score is calculated." Source: Business Insider, October 2023. [Link: https://www.businessinsider.com/personal-finance/what-is-fico-score].

[2]An oracle-efficient algorithm is one that calls a given oracle only polynomially many times.

a natural condition on the agents' cost function for which we give an oracle-efficient *approximation* algorithm for the best response of the agents for *any* hypothesis class.

- In Section 4, we study the learner's optimal information release problem. We consider screening/classification settings where agents are represented to the learner by a real-valued number that measures their qualification level for a certain task. Prior work has focused on similar one-dimensional settings in the context of strategic classification; see, e.g., [Beyhaghi et al., 2023, Braverman and Garg, 2020]. We first show that the learner's optimal information release problem is NP-hard when the agents' prior can be arbitrary. In light of this hardness result, we focus on *uniform* prior distributions and provide closed-form solutions for the case of *continuous* uniform priors, and an efficient algorithm to compute the optimal information release for *discrete* uniform priors.

- We finally consider alternative utility functions that are based on false positive (or negative) rates for the learner and provide insights as to what optimal information release should look under these utility functions, without restricting ourselves to uniform priors.

**Related Work.** Strategic classification was first formalized by Brückner and Scheffer [2011], Hardt et al. [2016]. Hardt et al. [2016] is perhaps the most seminal work in the area of strategic classification: they provide the first computationally efficient algorithms (under assumptions on the agents' cost function) to efficiently learn a near-optimal classifier in strategic settings. Importantly, this work makes the assumption that the agents fully know the exact parameters of the classifier due to existing "information leakage", even when the firm is obscuring their model. Hardt et al. [2016] also do not consider a learner that can release partial information about their model.

Closest to our work, Jagadeesan et al. [2021], Ghalme et al. [2021], Bechavod et al. [2022], and Haghtalab et al. [2023] relax the full information assumption and characterize the impact of opacity on the utility of the learner and agents. Jagadeesan et al. [2021] are the first to introduce a model of "biased" information about the learner's classifier: instead of observing the learner's deployed classifier exactly, agents observe and best respond to a noisy version of this classifier; one that is randomly shifted (by an additive amount) from the true deployed classifier.

In contrast, Ghalme et al. [2021] and Bechavod et al. [2022] consider models of *partial* information on the classifiers, where agents can access samples in the form of historical (feature vector, learner's prediction) pairs. More precisely, Ghalme et al. [2021] study what they coin the "price of opacity" in strategic classification, defined as the difference in prediction error when not releasing vs fully releasing the classifier. They are the first to show that this price can be positive (in the context of strategic classification), meaning that a learner can reduce their prediction error by fully releasing their classifier in strategic settings. Our work considers more general, intermediate forms of information release, instead of the all-or-nothing, binary approach of Ghalme et al. [2021].

Bechavod et al. [2022] consider a strategic regression setting in which the learner does not release their regression rule, but agents have access to (feature, score) samples as described above. They study how disparity in sample access (e.g., agents may only access samples from people similar to them) about the classifier across different groups induce unfairness in classification outcomes across these groups. Haghtalab et al. [2023] consider agents with (calibrated) forecasts over the actions of the learner, but do not consider the learner's information release which is our focus. Additionally, in our model, we do not constrain the agent's prior distribution to be calibrated.

Beyond strategic classification, there are a few related lines of work where such partial information is considered. One is Bayesian Persuasion [Kamenica and Gentzkow, 2011]: in Bayesian persuasion, the state of the world is randomly drawn from the prior, and there is a mapping from the state of the world to signal distributions. This mapping, i.e. the "signaling scheme", must be revealed to the agents in addition to the signal. In our setting, there is a *fixed* state of the world (the learner's classifier), and there is no need for the signaling scheme to be known, since the signal itself (the subset) reveals all the information needed for the agents. The agents only need to know that the learner is truthful, which is an assumption made in Bayesian persuasion too.

Relatedly, algorithmic recourse studies an "intermediate" information release problem where the learner publishes a recommended action or recourse for each agent to take, rather than a set of potential classifiers used by the learner; e.g., Harris et al. [2022]. In our model, we release the same signal or information to all agents based on the underlying distribution over these agents' features.

## 2 Model

Our model consists of a population of *agents* and a *learner*. Each agent in our model is represented by a pair $(x, y)$ where $x \in \mathcal{X}$ is a feature vector, and $y \in \{0, 1\}$ is a binary label. Throughout, we call an agent with $y = 0$ a "negative", and an agent with $y = 1$ a "positive". We assume there exists a mapping $f : \mathcal{X} \to \{0, 1\}$ that governs the relationship between $x$ and $y$; i.e., $y = f(x)$ for every agent $(x, y)$. We will therefore use $x$ to denote agents from now on. We denote by $D$ the distribution over the space of agents $\mathcal{X}$. Agent manipulations are characterized by a cost function $c : \mathcal{X} \times \mathcal{X} \to [0, \infty)$ where $c(x, x')$ denotes the cost that an agent incurs when changing their features from $x$ to $x'$. We assume, similar to standard strategic classification settings, that manipulation does *not* change one's true label: manipulation is seen purely as "gaming"; it does not change the qualification of an agent. Let $\mathcal{H} \subseteq \{0, 1\}^{\mathcal{X}}$ denote our hypothesis class, and let $h \in \mathcal{H}$ be the *fixed* classifier that the learner is using for classification.

**A Partial Knowledge Model for the Agents.** We move away from the standard assumption that agents fully know $h$ and model agents as having a *common* (shared by all agents) *prior distribution* $\pi$ over $\mathcal{H}$. This distribution captures their *initial* belief about which classifier is deployed by the learner. Formally, for every $h' \in \mathcal{H}$, $\pi(h')$ is the probability that the learner is going to deploy $h'$ for classification *from the agents' perspective.* We emphasize that the learner is committed to using a fixed classifier $h$. The prior $\pi$ captures the agents' belief about the deployed classifier and is known to the learner.

For example, job seekers may use Glassdoor to prepare for interviews. They may not know the exact hiring algorithm ($h$) of a specific company but can observe patterns from other companies for similar roles. This forms their initial belief, represented by $\pi$, about the classifier a company might use. Thus, $\pi$ captures the agents' probabilistic beliefs rather than assuming full knowledge of $h$.[3]

**A Partial Information Release Model for the Learner.** The learner has the ability to influence the agents' prior belief $\pi$ about the deployed classifier $h$ by releasing partial information about $h$. We model information release by releasing a subset $H \subseteq \mathcal{H}$ such that $h \in H$. We note that we reveal information truthfully, meaning that the deployed classifier is required to be in $H$.

Note that this is a general form of information release because it allows the learner to release *any* subset of the hypothesis class, so long as it includes the deployed classifier $h$. Below, we provide natural examples of information release that can be captured by our model.

**Example 2.1** (Examples of Information Release via Subsets). Consider the class of linear halfspaces in $d$ dimensions: $\mathcal{H} = \{h_{w,b} : w = [w^1, w^2, \ldots, w^d]^\top \in \mathbb{R}^d_+, b \in \mathbb{R}\}$ where $h_{w,b}(x) \triangleq \mathbb{1}[w^\top x + b \geq 0]$ and $x \in \mathcal{X} = \mathbb{R}^d$ is the feature vector. Let $h = h_{w_0, b_0}$ be the classifier deployed by the learner for some $w_0, b_0$. Under this setting, revealing the corresponding parameter of a feature, say $x^j$, in $h$ corresponds to releasing $H_1 = \{h_{w,b} \in \mathcal{H} : w^j = w_0^j\}$ (e.g., 'minimal GPA of 3.8 for grad school'). Revealing the top $k$ features of $h$ (e.g., the most significant class grades are algorithms and calculus) corresponds to releasing $H_2 = \{h_{w,b} \in \mathcal{H} : w^{i_1}, w^{i_2} \ldots, w^{i_k}$ are the $k$ largest coordinates of $w\}$. Let $I_0$ be such that $w_0^i \neq 0$ iff $i \in I_0$. Revealing the relevant features of $h$, i.e. features with nonzero coefficients (e.g., sensitive attributes like race or gender will not be used in the decision) corresponds to releasing $H_3 = \{h_{w,b} \in \mathcal{H} : w^i \neq 0, \forall i \in I_0\}$. This is a common form of information release in the real world[4].

**The Strategic Game with Partial Information Release.** Once the partial information $H$ is released by the learner, agents best respond as follows: each agent first computes their *posterior* belief about the deployed classifier by projecting their prior $\pi$ onto $H$, which we denote by $\pi|_H$, and is formally defined by: $\forall h' \in \mathcal{H}$, $\pi|_H(h') \triangleq \frac{\pi(h')}{\pi(H)} \mathbb{1}[h' \in H]$. Given this posterior distribution, the agent then moves to a new point that maximizes their utility. The utility is *quasi-linear* and measured by the

---

[3]Agent priors may also arise from observing previous decisions made by this classifier, for example as is studied in Bechavod et al. [2022]: the learner (e.g., a hiring company) has been using a classifier $h$ (the hiring algorithm) to screen agents (applicants) for some time. Agents going up for a decision today may observe some of the previous decisions made by the current classifier $h$ and use this information to form their prior.

[4]For example, recently, many U.S. universities have announced that they will not use race and ethnicity anymore in admissions, in line with a recent Supreme Court ruling.

|       | $h_1(= f)$ | $h_2$ | $h_3$ |
|-------|:----------:|:-----:|:-----:|
| $x_1$ |     1      |   0   |   0   |
| $x_2$ |     0      |   1   |   0   |

Table 1: Hypothesis class $\mathcal{H}$ in Example 2.3

probability (according to $\pi|_H$) of receiving a positive outcome minus the manipulation cost. Formally, the utility of agent $x$ that manipulates to $x'$, under the partial information $H$ released by the learner is given by

$$u_x(x', H) \triangleq \Pr_{h' \sim \pi|_H} [h'(x') = 1] - c(x, x'). \tag{1}$$

We let $\mathrm{BR}(x, H)$ denote the best response of agent $x$, i.e. a point $x'$ that maximizes $u_x(x', H)$. [5] The goal of the learner is to release $H$ that includes its deployed classifier $h$ so as to maximize its utility which is measured by its expected strategic accuracy.

$$U(H) \triangleq \Pr_{x \sim D} [h(\mathrm{BR}(x, H)) = f(x)]. \tag{2}$$

**Definition 2.2** (Strategic Game with Partial Information Release). *The game, between the learner who is using $h \in \mathcal{H}$ for classification, and the agents who have a prior $\pi$ over $\mathcal{H}$, proceeds as follows:*

1. *The learner (knowing $f$, $D$, $c$, $\pi$) publishes a subset of hypotheses $H \subseteq \mathcal{H}$ such that $h \in H$.*

2. *Every agent $x$ best responds by moving to a point $BR(x, H)$ that maximizes their utility: $BR(x, H) \in \mathrm{argmax}_{x' \in \mathcal{X}} u_x(x', H)$.*

*The learner's goal is to find a subset $H^\star \subseteq \mathcal{H}$ with $h \in H^\star$, that maximizes its utility[6]: $H^\star \in \mathrm{argmax}_{H \subseteq \mathcal{H}, h \in H} U(H)$*

We note that similar to standard strategic classification, the game defined in Definition 2.2 can be seen as a *Stackelberg* game in which the learner, as the "leader", commits to her strategy first and then the agents, as the "followers", respond. The optimal strategy of the learner, $H^\star$, corresponds to the *Stackelberg equilibrium* of the game, assuming best response of the agents.

**Contrasting with the Standard Setting of Strategic Classification.** The game defined in Definition 2.2 not only captures both the partial knowledge of the agents and the leaner's partial information release, but can also be viewed as a *generalization* of the standard strategic classification game where the agents fully observe the classifier $h$, which we refer to as the *full information release* game (e.g., see [Hardt et al., 2016]). This is because the learner can always choose $H = \{h\}$. Next, we ask:

*Can partial information release increase the learner's utility compared to full information release?*

Observe that by definition, $U(H^\star) \geq U(\{h\})$, i.e., the learner can only gain utility when they optimally release partial information instead of fully revealing the classifier. In the following examples, we show that there exist instantiations of the problem where $U(H^\star) > U(\{h\})$, even when $h$ is picked to be the optimal classifier in the full information release game, i.e., one that maximizes $U(\{h\})$. In other words, we show that the learner can gain *nonzero* utility by releasing a subset that is not $\{h\}$, *even if the choice of $h$ is optimized for the full information release game*.

**Example 2.3** (Partial vs. Full Information Release). Suppose $\mathcal{X} = \{x_1, x_2\}$, and that their probability weights under the distribution[7] are given by $D(x_1) = 2/3$, $D(x_2) = 1/3$, and their true labels are given by $f(x_1) = 1$, $f(x_2) = 0$. Suppose the cost function is given as follows: $c(x_1, x_2) = 2$, $c(x_2, x_1) = 3/4$. Let $\mathcal{H} = \{h_1, h_2, h_3\}$ be given by table 1. One can show that under this setting, $h = h_1$ is the optimal classifier under full information release, i.e., it optimizes $U(\{h\})$, and that for such $h$, $U(\{h\}) = 2/3$. However, suppose the prior distribution over $\mathcal{H}$ is uniform. One can show that under this setting, and when $h = h_1$ is the deployed classifier, releasing $H^\star = \{h_1, h_2\}$ implies $U(H^\star) = 1 > U(\{h\}) = 2/3$. In other words, the learner can exploit the agent's prior by releasing information in a way that increases its own utility by a significant amount.

---

[5]Ties are broken in favor of an arbitrarily lowest cost solution.

[6]We emphasize that here $h$ is fixed – namely, $H$ is the only variable in the optimization problem of the learner which is constrained to include $h$.

[7]The claim holds for any distribution $D$ in which both $x_1$ and $x_2$ are in the support.

---
**Algorithm 1:** Oracle$(c, \mathcal{H})$

---
**Input:** agent $x$, region $R = R^+ \cap R^-$ specified as, $R^+ = \cap_{i \in I^+} \{z : h_i(z) = 1\}$ and
       $R^- = \cap_{i \in I^-} \{z : h_i(z) = 0\}$ for some $I^+$ and $I^-$.
**Output:** $\text{argmin}_{z \in R}\, c(x, z)$

---

In the next example, we consider the more natural setting of single-sided threshold functions in one dimension and show that the same phenomenon occurs: the optimal utility achieved by partial information release is strictly larger than the utility achieved by the full information release of $h$, *even after the choice of $h$ is optimized for full information release*.

**Example 2.4** (Partial vs. Full Information Release). Suppose $\mathcal{X} = [0, 2]$, $D$ is the uniform distribution over $[0, 2]$, $f(x) = \mathbb{1}[x \geq 1.9]$, $\mathcal{H} = \{h_t : t \in [0, 2]\}$ where $h_t(x) \triangleq \mathbb{1}[x \geq t]$. Suppose the cost function is given by the distance $c(x, x') = |x - x'|$. We have that under this setting, the optimal classifier in $\mathcal{H}$ under full information release is $h = h_2$, and that its corresponding utility is $U(\{h\}) = 1 - \text{Pr}_{x \sim Unif[0,2]}[1 \leq x < 1.9] = 0.55$. Now suppose the agents have the following prior over $\mathcal{H}$: $\pi(h') = 0.1 \cdot \mathbb{1}[h' = h_2] + 0.9 \cdot \mathbb{1}[h' = h_{1.8}]$. Under this setting, and when $h = h_2$ is deployed for classification, one can see that releasing $H^\star = \{h_2, h_{1.8}\}$ leads to perfect utility for the learner. We therefore have $U(H^\star) = 1 > U(\{h\}) = 0.55$.

## 3    The Agents' Best Response Problem

In this section we focus on the best response problem faced by the agents in our model, as described in Definition 2.2. We consider a natural optimization oracle for the cost function of the agents that can solve simple projections. We will formally define this oracle later on. Given access to such an oracle, we then study the *oracle complexity*[8] of the agent's best response problem. First, we show that the best response problem is computationally hard even for a common family of $\ell_p$-norm cost functions. Next, we provide an *oracle-efficient* algorithm[9] for solving the best response problem when the hypothesis class is the class of low-dimensional linear classifiers. In Appendix B, we consider *submodular cost functions* and show that for any hypothesis class, there exists an oracle-efficient algorithm that *approximately* solves the best response problem in this setting.

Recall that the agents' best response problem can be cast as the following: given an agent $x \in \mathcal{X}$, and a distribution $P$ (e.g., $P = \pi|_H$ where $\pi$ is the prior and $H$ is the released information) over a set $\{h_1, \ldots, h_n\} \subseteq \mathcal{H}$, we want to solve $\text{argmax}_{z \in \mathcal{X}} \{\text{Pr}_{h' \sim P}[h'(z) = 1] - c(x, z)\}$. We consider an oracle that given any region $R \subseteq \mathcal{X}$, specified by the intersection of positive (or negative) regions of $h_i$'s, returns the projection of the agent $x$ onto $R$ according to the cost function $c$: $\text{argmin}_{z \in R}\, c(x, z)$. For example, when $\mathcal{H}$ is the class of linear classifiers and $c(x, z) = \|x - z\|_2$, the oracle can compute the $\ell_2$-projection of the agent $x$ onto the intersection of any subset of the linear halfspaces in $\{h_1, \ldots, h_n\}$. We denote this oracle by Oracle$(c, \mathcal{H})$ and formally define it in Algorithm 1.

Having access to such an oracle, and without further assumptions, the best response problem can be solved by exhaustively searching over all subsets of $\{h_1, \ldots, h_n\}$ because:

$$\max_{z \in \mathcal{X}} \left\{ \Pr_{h' \sim P}[h'(z) = 1] - c(x, z) \right\} = \max_{S \subseteq \{h_1, \ldots, h_n\}} \left\{ \sum_{h' \in S} P(h') - \min_{z : h'(z) = 1, \forall h' \in S} c(x, z) \right\} \quad (3)$$

This algorithm is inefficient because it makes exponentially many oracle calls. In what follows, we consider natural instantiations of our model and examine if we can get algorithms that make only $poly(n)$ oracle calls. All missing proof of this sections are provided in Appendix C.

**$p$-Norm Cost Functions.** First, we consider Euclidean spaces and the common family of $p$-norm functions for $p \geq 1$ and show that even under the assumption that the cost function of the agent belongs to this family, the problem of finding the best response is computationally hard. Formally, a $p$-norm cost function is defined by: for every $x, x' \in \mathbb{R}^d$, $c_p(x, x') = \|x - x'\|_p$ where $p \geq 1$.

**Theorem 3.1.** $\Omega(2^n/\sqrt{n})$ *calls to the oracle (Algorithm 1) are required to compute the best response of an agent with $2/3$ probability of success, even when $\mathcal{X} = \mathbb{R}^2$ and the cost function is $c_p$ for $p \geq 1$.*

---

[8]The number of times the oracle is called by an algorithm.

[9]An algorithm that calls the oracle only polynomially many times.

---

**Algorithm 2:** Best Response of Agents in the Linear Case

---

**Input:** agent $x$, cost function $c$, arbitrary distribution $P$ over linear classifiers $\{h_1, \ldots, h_n\}$
**Step 1. Compute the partitioning $(R_n)$ of the space given by** $\{h_1, \ldots, h_n\}$;
Initialize $R_1 \leftarrow \{\{z : h_1(z) = 1\}, \{z : h_1(z) = 0\}\}$;
**for** $i = 2, \ldots, n$ **do**
    $R_i \leftarrow R_{i-1}$;
    **for** $R \in R_{i-1}$ **do**
        **if** $\{z : h_i(z) = 0\} \cap R \neq \emptyset$ **then**
            $R_i \leftarrow R_i \setminus R$ ;                               `// Remove R`
            $R_i \leftarrow R_i \cup \{\{z : h_i(z) = 1\} \cap R, \{z : h_i(z) = 0\} \cap R\}$ ;       `// Split R`
**Step 2. Given $R_n$, compute the best response**;
**for** $R \in R_n$ **do**
    Let $R = R^+ \cap R^-$ where $R^+ = \cap_{i \in I^+} \{z : h_i(z) = 1\}$ and $R^- = \cap_{i \in I^-} \{z : h_i(z) = 0\}$;
    Call the oracle (Algorithm 1) to solve $z_R \in \operatorname{argmin}_{z \in R} c(x, z)$;
    Compute the utility of $z_R$: utility$(z_R) = \sum_{i \in I^+} P(h_i) - c(x, z_R)$;
**Output:** $\operatorname{argmax}_{z \in Z}$ utility$(z)$ where $Z = \{z_R : R \in R_n\}$

---

**Low-Dimensional Linear Classifiers.** Next, we show that when $\mathcal{X} = \mathbb{R}^d$ for some $d$, and when $\mathcal{H}$ contains only linear classifiers, i.e., every $h \in \mathcal{H}$ can be written as $h(x) = \mathbb{1}\left[w^\top x + b \geq 0\right]$ for some $w \in \mathbb{R}^d$ and $b \in \mathbb{R}$, then the best response of the agents can be computed with $O(n^d)$ oracle calls when $d \ll n$.

The algorithm, which is described in Algorithm 2, first computes the partitioning $(R_n)$ of the space $\mathcal{X}$ given by the $n$ linear classifiers. For any element of the partition in $R_n$, it then solves the best response when we restrict the agent to select its best response from that particular element. This gives us a set of points, one for each element of the partition. The algorithm finally outputs the point that has maximum utility for the agent. This point, by construction, is the best response of the agent. The oracle-efficiency of the algorithm follows from the observation that $n$ linear halfspaces in $d$ dimensions partition the space into at most $O(n^d)$ elements when $d \ll n$. Formally,

**Theorem 3.2.** *Suppose $\mathcal{X} = \mathbb{R}^d$ for some $d \ll n$, and $\mathcal{H}$ contains only linear classifiers. Then for any agent $x$, any cost function $c$, and any distribution $P$ over $\{h_1, \ldots, h_n\} \subseteq \mathcal{H}$, Algorithm 2 returns the best response of the agent in time $O(n^{d+1})$, while making $O(n^d)$ calls to the oracle (Algorithm 1).*

### 3.1 Generalizing to Arbitrary $P$

In Theorem 3.2 we require the distribution $P$ be over $\{h_1, \ldots h_n\} \subseteq \mathcal{H}$, e.g. to have finite support.

When this does not hold, ie. $P$ has infinite support size, we can ignore classifiers with sufficiently small probabilities (i.e., $poly(\epsilon)$), as they do not affect the manipulation strategy when searching for an $(1 + \epsilon)$-approximate solution. The number of classifiers in the support with probability at $poly(\epsilon)$ for a fixed $\epsilon > 0$ is at most $1/poly(\epsilon)$ which is a finite number. Therefore, to obtain a nearly optimal solution, it suffices to only consider probability distributions with finite support size.

## 4 The Learner's Optimal Information Release Problem

In this section we focus on the learner's optimization problem as described in Definition 2.2. The learner is facing a population of agents with prior $\pi$ and wants to release partial information $H \subseteq \mathcal{H}$ so as to maximize its utility $U(H)$. We note that the learner's strategy space can be restricted to the support of the agents' prior $\pi$ because including anything in $H$ that is outside of $\pi$'s support does not impact $U(H)$. Therefore, one naive algorithm to compute the utility maximizing solution for the learner is to evaluate the utility on all subsets $H \subseteq \text{support}(\pi)$ and output the one that maximizes the utility. However, this solution is inefficient; instead, can we have computationally efficient algorithms? We provide both positive and negative results for a natural instantiation of our model which is introduced below.

**The Setup: Classification Based on Test Scores.** We adopt the following setup for the learner's information release problem. We are motivated by screening problems such as school admissions and hiring where an individual's qualification level can be captured via a real-valued number, say, a test

score. We therefore consider agents that live in the one dimensional Euclidean space: $\mathcal{X} = [0, B] \subseteq \mathbb{R}$ for some $B$. One can think of each $x$ as the corresponding qualification level or test score of an agent where larger values of $x$ correspond to higher qualification levels or higher test scores. As we are in a strategic setting, agents can modify their true feature $x$ and "game" the learner by appearing more qualified than they actually are.

We let $f(x) = \mathbb{1}[x \geq t]$ for some $t$: there exists some threshold $t$ that separates qualified and unqualified agents. We take the hypothesis class $\mathcal{H}$ to be the class of all single-sided threshold classifiers: every $h' \in \mathcal{H}$ can be written as $h'(x) \triangleq \mathbb{1}[x \geq t']$ for some $t'$. We further take the cost function of the agents to be the standard distance metric in $\mathbb{R}$: $c(x, x') = |x' - x|$.[10]

**Remark 4.1.** *We emphasize that considering agents in the one-dimensional Euclidean space is only for simplicity of exposition. We basically assume, for an* arbitrary *space of agents $\mathcal{X}$, there exists a function $g : \mathcal{X} \to [0, B]$ such that $f(x) = \mathbb{1}[g(x) \geq t]$ for some $t$, and that the cost function is given by $c(x, z) = |g(z) - g(x)|$. Here, $g(x)$ captures the qualification level or test score of an agent $x$. Now observe that we can reduce this setting to the introduced setup of this section: take $\mathcal{X}' = \{g(x) : x \in \mathcal{X}\} \subseteq [0, B]$, $f : \mathcal{X}' \to \{0, 1\}$ is given by $f(x') = \mathbb{1}[x' \geq t]$, and that the cost function $c : \mathcal{X}' \times \mathcal{X}' \to [0, \infty)$ is given by $c(x', z') = |z' - x'|$.*

**Remark 4.2.** *Note that because every classifier $h' \in \mathcal{H}$ is uniquely specified by a real-valued threshold, for simplicity of our notations, we abuse notation and use $h'$ interchangeably as both a* mapping *(the classifier) and a* real-valued number *(the corresponding threshold) throughout this section. The same abuse of notation applies to $f$ as well.*

The classifier deployed by the learner is some $h \geq f$. We note that it is natural to assume $h \geq f$ because in our setup, higher values of $x$ are considered "better". So given the strategic behavior of the agents, the learner only wants to make the classification task "harder" compared to the ground truth $f$ — choosing $h < f$ will only hurt the learner's utility. Because we will extensively make use of the fact that $h \geq f$, we state it as a remark below.

**Remark 4.3.** *The learner's adopted classifier is some $h \in \mathcal{H}$ such that $h \geq f$.*

First, we show that under the introduced setup, the learner's optimization problem is NP-hard if the prior can be chosen arbitrarily. This is shown by a reduction from the NP-hard *subset sum* problem. The formal NP-hardness statement and its proof, as well as further useful facts about the agents' best response under this setup are in Appendix D.

## 4.1 An Efficient Algorithm for Discrete Uniform Priors

Given the hardness of the learner's problem for arbitrary prior distributions, we focus on a specific family of priors, namely, *uniform* priors over a given set, and examine the existence of efficient algorithms for such priors. In Appendix D.2, we consider *continuous* uniform priors and provide closed form solutions for the learner's optimal partial information release problem.

In this section, we provide an efficient algorithm for computing the learner's optimal information release when the prior $\pi$ is a *discrete* uniform distribution over a set $\{h_1, h_2, \ldots, h_n\} \subseteq \mathcal{H}$ that includes the adopted classifier $h$. Here, the objective of the learner is to release a subset $H \subseteq \{h_1, h_2, \ldots, h_n\}$ such that $h \in H$. Throughout, we take $h = h_k$ where $1 \leq k \leq n$, and assume $h_1 \leq h_2 \leq \ldots \leq h_n$. The complete exposition of this section, including all details, proofs, necessary lemmas, and the description of the proposed algorithm, can be found in Appendix E.

We first characterize the utility of any subset $H$ released by the learner using a real-valued function of $H$. Define, for any $H \subseteq \{h_1, \ldots, h_n\}$ such that $h \in H$,

$$R_H \triangleq \inf \{x : \mathrm{BR}(x, H) \geq h\} \tag{4}$$

Note that $\mathrm{BR}(x = h, H) \geq h$ for any $H$ such that $h \in H$. Therefore, $\{x : \mathrm{BR}(x, H) \geq h\}$ is nonempty, and that $R_H \leq h$ for any $H$ such that $h \in H$. In the following lemma, we show that $R_H$ characterizes the utility of $H$ for the learner, for any prior $\pi$ over $\{h_1, \ldots, h_n\}$.

**Lemma 4.4.** *Fix any prior $\pi$ over $\{h_1, \ldots, h_n\}$. We have that for any $H \subseteq \{h_1, \ldots, h_n\}$ such that $h \in H$, the utility of the learner, given by Equation 2, can be written as*

$$U(H) = \begin{cases} 1 - \mathrm{Pr}_{x \sim D}[R_H < x < f] & R_H < f \\ 1 - \mathrm{Pr}_{x \sim D}[f \leq x \leq R_H] & R_H \geq f \end{cases} \tag{5}$$

---

[10]Our results can be extended to the case where $c(x, x') = k|x' - x|$ for some constant $k$.

Given such characterization of the learner's utility by $R_H$, we show that when the agents' prior is uniform over $\{h_1, \ldots, h_n\}$, there are only *polynomially many* possible values that $R_H$ can take, even though *there are exponentially many $H$'s*. We characterize the set of possible values for $R_H$ as well. For any possible value $R$ of $R_H$, our algorithm efficiently finds a subset $H$ such that $R_H = R$, if such $H$ exists, and finally outputs the one with maximal utility.

More formally, we consider the following partitioning of the space of subsets of $\{h_1, \ldots, h_n\}$. For any $\ell \in \{1, 2, \ldots, n\}$, and for any $i \in \{k, k+1, \ldots, n\}$, define

$$S_{i,\ell} \triangleq \{H \subseteq \{h_1, \ldots, h_n\} : h \in H, |H| = \ell, \text{BR}(h, H) = h_i\}$$

Note that $\text{BR}(h \equiv h_k, H) \in \{h_i : i \geq k\}$ for any $H$. Therefore, $\{S_{i,\ell}\}_{i,\ell}$ gives us a proper partitioning of the space of subsets, which implies

$$\max_{H \subseteq \{h_1, \ldots, h_n\}, h \in H} U(H) = \max_{i,\ell} \max_{H \in S_{i,\ell}} U(H)$$

Given this partitioning of the space, we show that $R_H$ can be characterized as follows.

**Lemma 4.5.** *If the prior $\pi$ is uniform over $\{h_1, \ldots, h_n\}$, then for any $H \in S_{i,l}$, $R_H = h_i - j/\ell$ where $j = |\{h' \in H : h' \in (R_H, h_i]\}|$.*

Given such characterization, our proposed algorithm (Algorithm 3), for any $i, \ell$, enumerates over all possible $j \in \{1, \ldots, \ell\}$ and returns a $H$ such that $H \in S_{i,\ell}$ and $R_H = h_i - j/\ell$, if such $H$ exists. The algorithm then outputs the subset $H$ with maximal utility according to Equation 5.

**Theorem 4.6.** *There exists an algorithm (Algorithm 3) that for any $n$, any uniform prior over $\{h_1, \ldots, h_n\}$ that includes $h$, and any data distribution $D$, returns $H^\star \subseteq \{h_1, \ldots, h_n\}$ in time $O(n^3)$ such that $h \in H^\star$, and that $U(H^\star) = \max_{H \subseteq \{h_1, \ldots, h_n\}, h \in H} U(H)$.*

### 4.2 Minimizing False Positive (Negative) Rates for Arbitrary Priors

While so far we worked with *accuracy* as the utility function of the learner, in this section, we consider other natural performance metrics and provide insights on the optimal information release for the proposed utility functions, without restricting ourselves to uniform priors. In particular, we consider utility functions that are based on *False Negative Rate* (FNR) and *False Positive Rate* (FPR) which are formally defined below. For any $H \subseteq \mathcal{H}$ such that $h \in H$,

$$U_{FPR}(H) \triangleq 1 - FPR(H) \triangleq 1 - \Pr_{x \sim D}[h(BR(x, H)) = 1 | f(x) = 0] \tag{6}$$

$$U_{FNR}(H) \triangleq 1 - FNR(H) \triangleq 1 - \Pr_{x \sim D}[h(BR(x, H)) = 0 | f(x) = 1] \tag{7}$$

In the following theorem, we establish that for any given prior $\pi$ over a set $\{h_1, h_2, \ldots, h_n\} \subseteq \mathcal{H}$, if the learner aims to minimize the FPR, *no-information-release* is preferable to *full-information-release*.[11] Additionally, we show that for minimizing the FNR, an optimal strategy for the learner is *full-information-release*. By "no-information-release", we mean releasing any subset $H$ such that $H$ includes the support of the prior $\pi$: $H \supseteq \{h_1, \ldots, h_n\}$ which results in $\pi|_H = \pi$. By "full-information-release", we mean revealing the classifier: $H = \{h\}$.

**Theorem 4.7.** *Fix any $h \geq f$. For any prior $\pi$ over $\{h_1, \ldots, h_n\}$ that includes $h$, we have 1) $FPR(\mathcal{H}) \leq FPR(\{h\})$. 2) $FNR(\{h\}) \leq FNR(H)$ for every $H \subseteq \mathcal{H}$ such that $h \in H$.*

The proof is provided in Appendix F. In Appendix G, we show that minimizing FPR, unlike minimizing FNR, does not always have a clear optimal solution. We provide three instances such that full-information-release is optimal for the first, no-information-release is optimal for the second, and neither is optimal for the third.

## 5 Conclusion

We introduce *Bayesian Strategic Classification*, meaning strategic classification with partial knowledge (of the agents) and partial information release (of the learner). Our model relaxes the often

---

[11]We note that in this section, while we work with discrete priors over some $\{h_1, \ldots, h_n\} \subseteq \mathcal{H}$, our results can be easily extended to *any* prior.

unrealistic assumption that agents fully know the learner's deployed classifier. Instead, we model agents as having a distributional prior on which classifier the learner is using. Our results show the existence of previously unknown intriguing informational middle grounds; we also demonstrate the necessity of revisiting the fundamental modeling assumptions of strategic classification in order to provide effective recommendations to practitioners in high-stakes, real-world prediction tasks.

## Acknowledgements

The authors thank Avrim Blum for helpful discussions in the early stages of this work. Special thanks to Roy Long for suggesting that our model might leak more information than intended, and to Odelia Lorch for identifying an instance where this occurs and providing a proof (located in Appendix H).

Lee Cohen is supported by the Simons Foundation Collaboration on the Theory of Algorithmic Fairness, the Sloan Foundation Grant 2020-13941, and the Simons Foundation investigators award 689988. Kevin Stangl was supported in part by the National Science Foundation under grants CCF-2212968 and ECCS-2216899, by the Simons Foundation under the Simons Collaboration on the Theory of Algorithmic Fairness, and by the Defense Advanced Research Projects Agency under cooperative agreement HR00112020003. The views expressed in this work do not necessarily reflect the position or the policy of the Government and no official endorsement should be inferred.

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

# A    Limitations, and Broader Impacts, and Future Work

**Limitations**    Potential limitations of our model are the following:

- We assume the agents' prior is realizable, in the sense that the classifier deployed by the learner is in the support of the prior. This is a standard assumption in machine learning works, and it will be interesting to relax it in future works.

- The learner, in order to decide on the optimal level of information release, must know the agents' prior. This assumption, while common in related settings like Bayesian persuasion, may be unrealistic in practice; although, in real-life, the learner may have an imperfect idea of or be able to partially recover the agents' priors from previous interactions with them. Beyond this, we note that in practice, different agents may also have different beliefs and priors about the learner's model; this can affect the way the learner should release information, given that this information release may affect different users differently.

- A limitation of this model is that it assumes the learner must commit to a fixed classifier in advance. In real-life, classifiers are dynamically updated over time, using the additional information obtained from each decision. However, we note that changing the screening algorithm requires significant resources, and the rate at which the classifier is updated is generally slower than the rate at which decisions are made. In practice, this means that strategic agents effectively face a fixed model in each "batch" between updates.

- If agents know the prior distribution $D$ and the mapping $f$ from feature vectors to labels, they might infer information regarding $h^*$ when $|H| > 1$. We show an example of such a case in Appendix H.

**Broader Impacts**    On the plus side, our approach provides a deeper understanding of strategic behavior in machine learning settings, when strategic agents may not fully understand the deployed model. By doing so, we are taking the understanding of strategic classification one step closer to real life, providing useful insights on how much information a learner should provide about their model to prevent model manipulation and gaming.

One potential negative impact is that our approach takes the point of view of the learner who is solely interested in maximizing his own accuracy (or utility). It is well-known that this focus on accuracy can lead to unfairness and disparate harms across different populations; further, prior work studying fairness in the standard strategic classification setting [Hu et al., 2019, Milli et al., 2019] and in a related partial information setting [Bechavod et al., 2022] have shown that strategic classification can *amplify* these disparities.

**Future Work**    [Hu et al., 2019, Milli et al., 2019] consider disparities across different groups due to *differing cost functions*. In our model of strategic classification, different population groups may not only have differing cost functions but also *differing prior distributions*: network homophily, social disparities, and stratification can cause population groups to have distinct priors, leading to further disparities across groups. In turn, it will be critical in future work to design *fairness-aware* information release strategies by a learner faced with strategic behavior.

# B    Oracle-Efficient Approximate Best Response for $V$-Submodular Costs

In this section we give a sufficient condition on the cost function under which we can give an approximation algorithm for the best response of the agents. In particular, for a given collection of classifiers $V \subseteq \mathcal{H}$, we introduce the notion of $V$-*submodular* cost functions which is a natural condition that can arise in many applications. Borrowing results from the literature on submodular optimization, we then show that for any distribution $P$ over a set $V = \{h_1, \ldots, h_n\}$, if the cost function is $V$-submodular, there exists an oracle-efficient approximation algorithm for the best response problem. Recall, from Equation (3), that the best response problem faced by agent $x$ can be written as $\max_{S \subseteq \{h_1, \ldots, h_n\}} g_x(S) \triangleq \sum_{h' \in S} P(h') - c(x, S)$ where, with slight abuse of notation, we define

$$c(x, S) \triangleq \min_{z : h'(z) = 1, \, \forall h' \in S} c(x, z) \tag{8}$$

For any $S \subseteq \{h_1, \ldots, h_n\}$, $c(x, S)$ is simply the minimum cost that the agent $x$ has to incur in order to pass all classifiers in $S$, and can be computed via the oracle (Algorithm 1). We now state our main assumption on the cost function:

**Definition B.1** (*$V$-Submodularity*). *Let $V = \{h_1, \ldots, h_n\}$ be any collection of classifiers. We say a cost function $c$ is $V$-submodular, if for all $x$, the set function $c(x, \cdot) : 2^V \to \mathbb{R}$ defined in Equation 8 is submodular: for every $S, S' \subseteq V$ such that $S \subseteq S'$ and every $h' \notin S'$,*

$$c(x, S \cup \{h'\}) - c(x, S) \geq c(x, S' \cup \{h'\}) - c(x, S')$$

This condition asks that the marginal cost of passing the new classifier $h'$ is smaller when the new classifier is added to $S'$ versus $S$, for any such $h', S, S'$. Fix a collection of classifiers $V$. Informally speaking, a cost function is $V$-submodular if the agent's manipulation to pass a classifier only helps her (i.e., reduces her cost) to pass other classifiers: the more classifiers the agent passes, it becomes only easier for her to pass an additional classifier. This can happen in settings where some of the knowledge to pass a certain number of tests is *transferable* across tests. Some real-life examples include: 1) a student that is preparing for a series of math tests on topics like probability, statistics, and combinatorics. 2) a job applicant who is applying for multiple jobs within the same field and preparing for their interviews.

We give a formal example of a $V$-submodular cost function below. In particular, we show that when $\mathcal{X} = \mathbb{R}$, the cost function $c(x, x') = |x - x'|$ is $V$-submodular where $V$ can be any set of single-sided threshold classifiers.

**Claim B.2.** *Let $\mathcal{X} = \mathbb{R}$, and $V = \{h_1, \ldots, h_n\}$ where every $h_i$ can be written as $h_i(x) = \mathbb{1}[x \geq t_i]$ for some $t_i \in \mathbb{R}$. We have that the cost function $c(x, x') = |x - x'|$ is $V$-submodular.*

*Proof of Claim B.2.* We will abuse notation and use $h_i$ for the threshold $t_i$ ($h_i \equiv t_i \in \mathbb{R}$).

Fix any $x$. Consider $S \subseteq S' \subseteq \{h_1, \ldots, h_n\}$ and $h' \in \mathbb{R}$ such that $h' \notin S'$. Note that

$$c(x, S) = \max\left(\max(S) - x, 0\right), \quad c(x, S \cup \{h'\}) = \max\left(\max(S \cup \{h'\}) - x, 0\right)$$
$$c(x, S') = \max\left(\max(S') - x, 0\right), \quad c(x, S' \cup \{h'\}) = \max\left(\max(S' \cup \{h'\}) - x, 0\right)$$

where $\max(F)$ is simply the largest threshold in $F$, for any set $F$. Note that $\max(S) \leq \max(S')$ because $S \subseteq S'$. Suppose $x \leq \max(S)$. We have three cases

1. If $h' \geq \max(S')$, then

$$c(x, S \cup \{h'\}) - c(x, S) = h' - \max(S) \geq h' - \max(S') = c(x, S' \cup \{h'\}) - c(x, S')$$

2. If $\max(S) \leq h' \leq \max(S')$, then

$$c(x, S \cup \{h'\}) - c(x, S) = h' - \max(S) \geq 0 = c(x, S' \cup \{h'\}) - c(x, S')$$

3. If $h' \leq \max(S)$, then

$$c(x, S \cup \{h'\}) - c(x, S) = c(x, S' \cup \{h'\}) - c(x, S') = 0$$

So we have shown that the cost function is submodular if $x \leq \max(S)$. We can similarly, using a case analysis, show that the cost function is submodular when $x > \max(S)$. $\quad\square$

We now state the main result of this section.

**Theorem B.3.** *Fix any $\mathcal{H}$ and any distribution $P$ over some $V = \{h_1, \ldots, h_n\} \subseteq \mathcal{H}$. If the cost function $c$ is $V$-submodular, then there exists an algorithm that for every agent $x$ and every $\epsilon > 0$, makes $\tilde{O}(n/\epsilon^2)$ calls to the oracle (Algorithm 1) and outputs a set $\hat{S} \subseteq V$ such that $g_x(\hat{S}) \geq \max_{S \subseteq V} g_x(S) - \epsilon$.*

*Proof of Theorem B.3.* Note that when the cost function is $V$-submodular, the objective function $g_x$ can be written as the difference of a monotone non-negative modular function[12] and a monotone non-negative submodular function: $g_x : 2^V \to \mathbb{R}$, $g_x(S) = \sum_{h' \in S} P(h') - c(x, S)$. The result then follows from [El Halabi and Jegelka, 2020] where they provide an efficient algorithm for approximately maximizing set functions with such structure. $\quad\square$

---

[12]A set function $r$ is modular if $r(S) = \sum_{s \in S} r(s)$ for any $S$.

# C Missing Proofs of Section 3

**Theorem 3.1.** $\Omega(2^n/\sqrt{n})$ *calls to the oracle (Algorithm 1) are required to compute the best response of an agent with $2/3$ probability of success, even when $\mathcal{X} = \mathbb{R}^2$ and the cost function is $c_p$ for $p \geq 1$.*

*Proof of Theorem 3.1.* To prove the claim, we reduce the following hidden-set detection problem with EQUALTO$(\cdot)$ oracle to our best response problem. In hidden-set detection problem, given two players, Alice and Bob, with Bob possessing a 'hidden' subset $S^\star \subseteq [n]$ of size $n/2$, Alice aims to identify Bob's set $S^\star$ using the minimum number of queries to Bob. She has query access to EQUALTO$(T)$ oracle that checks whether her set $T \subset [n]$ matches Bob's set $(S^\star)$. It is trivial that any randomized algorithm for the hidden-set detection problem with success probability of at least $1 - O(1)$ requires $\binom{n}{n/2}$ queries in the worst-case scenario. This is via a straightforward application of Yao's Min-Max principle Yao [1977]: consider a uniform distribution over all subsets of size $n/2$ from $[n]$, as the Bob's set. Then, after querying half of the subsets of size $n/2$, the failure probability of Alice in detecting Bob's set is at least $(1-1/n)(1-1/(n-1))\cdots(1-1/(n/2)) > (1-2/n)^{n/2} > e^{-1}(1-2/n) > 1/3$ for sufficiently large values of $n$.

Next, corresponding to an instance of the hidden-set detection problem with $S^\star$, we create an instance of the agents' best response problem and show that any algorithm that computes the best response with success probability at least $2/3$ using $N$ oracle calls (Algorithm 1), detects the hidden set of the given instance of the hidden-set detection problem using at most $N$ calls of EQUALTO$(\cdot)$ with probability at least $2/3$. Hence, computing the best response problem with success probability at least $2/3$ requires $\binom{n}{n/2} = \Omega(2^n/\sqrt{n})$ oracle calls.

Let $n = 2k$ and $\epsilon < 1/n$. Corresponding to every subset $S \subset [n]$ of size $n/2 - 1$, there is a distinct point $x_S$ at distance $1/2 - \epsilon$ from the origin, i.e., $\|x_S\|_p = 1/2 - \epsilon$. Corresponding to every subset $S \subset [n]$ of size $n/2$, there are two distinct points $x_{S,n}$ and $x_{S,f}$ at distances respectively $1/2 - \epsilon$ (near) and $1/2 + \epsilon$ (far) from the origin, i.e., $\|x_{S,n}\|_p = 1/2 - \epsilon$ and $\|x_{S,f}\|_p = 1/2 + \epsilon$.

Now, we are ready to describe the instance $I_{S^\star}$ of our best response problem corresponding to the given hidden-set detection problem with $S^\star$. We define $\mathcal{H} = \{h_1, \cdots, h_n\}$ and distribution $P$ over $\mathcal{H}$ such that,

- $P$ is a uniform distribution over all classifiers $\mathcal{H}$, i.e., $P(h_i) = 1/n$ for every $i \in [n]$.

- For every subset $T \subset [n]$ of size $n/2 - 1$, we define $h_i(x_T) = \mathbb{1}[i \in T]$.

- For every subset $T \subset [n]$ of size $n/2$, we define $h_i(x_{T,f}) = \mathbb{1}[i \in T]$. Moreover, if $T \neq S^\star$, then $h_i(x_{T,n}) = 0$ for all $i \in [n]$. Otherwise, if $T = S^\star$, we define $h_i(x_{T,n}) = \mathbb{1}[i \in T]$.

- Finally, for the remaining points in $\mathcal{X}$, i.e., $x' \in \mathbb{R}^2 \setminus (\{x_T : T \subset [n] \text{ and } |T| = n/2 - 1\} \cup \{x_{T,n}, x_{T,f} : T \subset [n] \text{ and } |T| = n/2\})$, we define $h_i(x') = 0$ for all $i \in [n]$. In other words, points that do not correspond to subsets of size $n/2 - 1$ or $n/2$ are classified as negative examples by every classifier in $\mathcal{H}$.

In the constructed instance $I_{S^\star}$, no point is classified as positive by more than $n/2$ classifiers in $\mathcal{H}$, and the $p$-norm distance from the origin for all points classified as positive by a subset of classifiers is at least $1/2 - \epsilon$. Therefore, the best response for an agent located at the origin of the space is $x_{S^\star,n}$, yielding a utility of $1/2 - (1/2 - \epsilon) = \epsilon > 0$. Hence, the computational task in computing the best response involves identifying the (hidden) subset $S^\star$. Refer to Figure 1 for a description of $I_{S^\star}$.

Although we described the construction of $I_{S^\star}$, what we need to show to get the exponential lower bound on the oracle complexity of the best response problem is constructing an oracle (i.e., an implementation of Algorithm 1), using the EQUALTO$(\cdot)$ oracle, consistent with $I_{S^\star}$. To do so, given a subset of classifiers specified by $T \subset [n]$, the oracle returns as follows:

- **if** $|T| > n/2$: It returns an empty set.

- **if** $|T| < n/2$: It returns $x_{T'}$ for an arbitrary set $T' \supseteq T$ of size $n/2 - 1$. Note that $\|x_{T'}\|_p = 1/2 - \epsilon$.

- **if** $|T| = n/2$ **and** **EQUALTO**$(T) =$ **FALSE**: It returns $x_{T,f}$. Note that $\|x_{T,f}\|_p = 1/2 + \epsilon$.

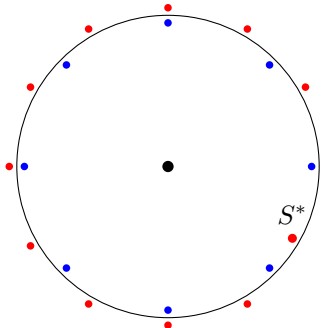

Figure 1: In this example, we consider $p = 2$, i.e., $c(x, x') = \|x, x'\|_2$. The agent is located at the origin. Blue nodes correspond to a point in the intersection of the positive regions of subsets of classifiers of size $\frac{n}{2} - 1$, each located at a Euclidean distance of $1/2 - \epsilon$ from the origin, where $\epsilon$ is a small positive value. Moreover, points in the intersection of the positive regions of subsets classifiers of size $\frac{n}{2}$ are indicated by red points, all except the one corresponding to $S^\star$ are located at a Euclidean distance of $1/2 + \epsilon$ from the origin. The red point corresponding to $S^\star$ is uniquely placed at a distance of $1/2 - \epsilon$ from the origin, similar to the blue nodes. Furthermore, all points, corresponding to different subsets, are located at distinct locations in the space.

- **if $|T| = n/2$ and EQUALTO$(T)$ = TRUE**: It returns $x_{T,n}$. Note that $\|x_{T,n}\|_p = 1/2 - \epsilon$.

$\square$

**Remark C.1.** *As in each instance $I_{S^\star}$ the only point with strictly positive utility is $x_{S^\star,n}$, our proof for Theorem 3.1 essentially rules out the existence of any approximation algorithm for the best response problem with success probability at least $2/3$ using $o(2^n/\sqrt{n})$.*

**Theorem 3.2.** *Suppose $\mathcal{X} = \mathbb{R}^d$ for some $d \ll n$, and $\mathcal{H}$ contains only linear classifiers. Then for any agent $x$, any cost function $c$, and any distribution $P$ over $\{h_1, \ldots, h_n\} \subseteq \mathcal{H}$, Algorithm 2 returns the best response of the agent in time $O(n^{d+1})$, while making $O(n^d)$ calls to the oracle (Algorithm 1).*

*Proof of Theorem 3.2.* The fact that Algorithm 2 returns the best response follows from the construction of the algorithm. We first prove the oracle and the runtime complexity for $d = 2$ and then generalize it to any $d$. The oracle complexity of Algorithm 2 is $|R_n|$. Note that $|R_1| = 2$, and for any $n \geq 2$, $|R_n| \leq |R_{n-1}| + n$. This is because the line $\{z : h_n(z) = 0\}$ intersects the lines formed by $\{h_1, \ldots, h_{n-1}\}$ in at most $n - 1$ points, which will then partition $\{z : h_n(z) = 0\}$ into at most $n$ segments. Each segment of the new line then splits a region in $R_{n-1}$ into two regions. So, there are at most $n$ new regions when $h_n$ is introduced. The recursive relationship implies that $|R_n| \leq 1 + \frac{n(n+1)}{2} = O(n^2)$. The runtime complexity of the algorithm is then given by $O\left(\sum_{i=1}^n |R_i|\right) = O(n^3)$.

Now consider any dimension $d$ and let $R(n, d)$ denote the number of partitions induced by the classifiers $\{h_1, \ldots, h_n\}$. Note that in this case we have $R(n, d) \leq R(n-1, d) + R(n-1, d-1)$. The first term on the right hand side is the number of regions induced by $\{h_1, \ldots, h_{n-1}\}$ and the second term is the number of splits (dividing a region into two) when $h_n$ is introduced. Note that $\{z : h_n(z) = 0\}$ is a $d-1$-dimensional hyperplane and the number of splits induced by $h_n$ is simply the number of regions induced by $\{h_1, \ldots, h_{n-1}\}$ on $\{z : h_n(z) = 0\}$, which is $R(n-1, d-1)$. The recursive relationship implies that $|R_n| = R(n, d) \leq \sum_{j=0}^d \binom{n}{j} = O(n^d)$. $\square$

## D Missing Details of Section 4

We first state a remark about the tie-breaking of agents' best response problem:

**Remark D.1.** *As mentioned in the model section, when there are several utility-maximizing solutions for the agents, we always break ties in favor of the lowest cost solution. Furthermore, each agent $x$ in our setup manipulate* only *to larger values of $x$ ($x' \geq x$); this is formally stated in the first part of*

*Lemma D.2. Therefore, the tie-breaking of agents is in favor of smaller values of $x'$ in our setup. In other words, given some released information $H$, an agent $x$ chooses*

$$BR(x, H) = \min \left\{ \operatorname*{argmax}_{x' \geq x} u_x(x', H) \right\} \tag{9}$$

*In the rest of this section, when we write $\operatorname{argmax}_{x' \geq x} u_x(x', H)$, we implicitly are taking the smallest $x' \geq x$ that maximizes the utility of the agent $x$.*

Next, we state some useful facts about the agents' best response in our setup.

**Lemma D.2.** *Fix any prior $\pi$ and any points $x_2 \geq x_1$. We have that, for any $H \subseteq \mathcal{H}$,*

1. *$BR(x_1, H) \geq x_1$.*

2. *$BR(x_2, H) \geq BR(x_1, H)$.*

3. *If $BR(x_1, H) \geq x_2$, then $BR(x_1, H) = BR(x_2, H)$.*

*Proof of Lemma D.2.* Fix any $x$, and any $H$. We have

$$\begin{aligned}
\mathrm{BR}(x, H) &= \operatorname*{argmax}_{x'} \left\{ \Pr_{h' \sim \pi|_H} [x' \geq h'] - |x' - x| \right\} \\
&= \operatorname*{argmax}_{x' \geq x} \left\{ \Pr_{h' \sim \pi|_H} [x' \geq h'] - (x' - x) \right\} \\
&= \operatorname*{argmax}_{x' \geq x} \left\{ \Pr_{h' \sim \pi|_H} [x' \geq h'] - x' \right\} \\
&= \operatorname*{argmax}_{x' \geq x} g_H(x')
\end{aligned}$$

where we take $g_H(x') \triangleq \Pr_{h' \sim P|_H} [x' \geq h'] - x'$. The first equality follows because agents do not gain any utility by moving to a point $x' < x$, and that tie-breaking is in favor of lowest cost solution.

The first and the second part of the lemma follows from this derivation. For the third part, we have

$$\mathrm{BR}(x_1, H) = \operatorname*{argmax}_{x' \geq x_1} g_H(x') = \operatorname*{argmax}_{x' \geq x_2} g_H(x') = \mathrm{BR}(x_2, H)$$

where the second equality follows because $\mathrm{BR}(x_1, H) \geq x_2$. $\qquad\square$

## D.1 NP-Hardness of Learner's Optimization Problem with Arbitrary Prior Distributions

In this section, we formally state the NP-hardness of the learner's optimization problem in the general setting.

**Theorem D.3.** *Consider an arbitrary prior $\pi$ over a set of threshold classifiers $\{h_1, h_2, \ldots, h_n\} \subseteq \mathcal{H}$ that includes $h$. The problem of finding $H \subseteq \{h_1, h_2, \ldots, h_n\}$ so that $h \in H$ and the learner's utility $U(H)$ is maximized is NP-hard.*

*Proof of Theorem D.3.* The proof is via a reduction from the subset sum problem. In particular, we consider a variant of the subset sum problem in which we are given a set of $n$ positive numbers $a_1, \cdots, a_n$, and the goal is to decide whether a subset $S \subset [n]$ such that $\sum_{i \in S} a_i = T := (1/2) \sum_{i \in [n]} a_i$ exists.

Given an instance of the subset sum problem with input $(\{a_1, \cdots, a_n\}, T := (1/2) \sum_{i \in [n]} a_i)$, we construct the following instance of our problem with one-dimensional threshold classifiers. Define $f(x) = \mathbb{1}[x \geq 0]$, $h(x) = \mathbb{1}[x \geq 2/3]$, and $h_i(x) = \mathbb{1}[x \geq 100 + i]$ for every $i \in [n]$. Moreover, suppose that the prior distribution of the agents $\pi$ is given by: $\pi(h) = 1/2$ and for every $i \in [n]$, $\pi(h_i) = a_i/(4T)$. Note that $\pi(h) + \sum_i \pi(h_i) = 1$. Let the data distribution $D$ be the uniform distribution over $[-1000, 1000]$.

Intuitively speaking, the inclusion of $h_i$'s in $H$ have no direct effect on the accuracy of the released subset $H$, as they can only lead to a subset of the agents located at $x \geq 100$ to manipulate. However,

their presence in $H$ will impact the probability mass of $h$ under the posterior $\pi|_H$, which is given by $\pi|_H(h) = \pi(h)/\pi(H) \triangleq \rho_H$. We will show that the learner can achieve perfect accuracy *if and only if* in the given instance subset sum problem there exists a subset which sums up to $T$. To see this consider the following cases for the released information $H$.

- **Case 1:** $\rho_H > 2/3$. For any such $H$, all agents at distance $\rho_H$ from $2/3$ gain positive utility by manipulating to $x' = 2/3$. Hence, the utility of such solutions for the learner is given by $1 - \Pr_{x \sim D}[x \in [\frac{2}{3} - \rho_H, 0)] < 1$.

- **Case 2:** $\rho_H < 2/3$. For any such $H$, as all classifiers in $H \setminus \{h\}$ are located at $t > 100$, no agent belonging to $[0, 2/3 - \rho_H)$ gain positive utility by manipulating to $x' = 2/3$. Hence, these points will be misclassified by $h$, and consequently, the utility of such solutions for the learner is given by $1 - \Pr_{x \sim D}[x \in [0, \frac{2}{3} - \rho_H)] < 1$.

- **Case 3:** $\rho_H = 2/3$. By similar arguments to the previous cases, all agents belonging to $[0, 2/3)$ manipulate to $x' = 2/3$ and all points with negative labels ($x < 0$) stay at their location. Therefore, no one will be misclassified, and therefore, the utility of such solutions is 1.

Note that because $\rho_H = \frac{\pi(h)}{\pi(H)} = \frac{1/2}{1/2 + \pi(H \cap \{h_1, \cdots, h_n\})}$, we have that $\rho_H = 2/3$ if and only if $\pi(H \cap \{h_1, \cdots, h_n\}) = 1/4$. But $\pi(H \cap \{h_1, \cdots, h_n\}) = 1/(4T) \sum_{h_i \in H} a_i$. We therefore have that $\rho_H = 2/3$ *if and only if* $\sum_{h_i \in H} a_i = T$. Hence, deciding whether the learner's optimization problem has a solution with perfect utility is equivalent to deciding whether in the given subset sum problem there exists a subset $S \subset [n]$ such that $\sum_{i \in S} a_i = T := (1/2) \sum_{i \in [n]} a_i$. $\qquad \square$

## D.2 A Closed-form Solution for Continuous Uniform Priors

In this section, we provide closed-form solutions for *continuous* uniform priors. More concretely, we assume in this section that $\pi$ is the uniform distribution over an interval $[a, b] \subset \mathbb{R}$ that includes $h$. The information release of the learner will then be releasing an interval $H = [c, d] \subseteq [a, b]$ such that $h \in [c, d]$.

For example, a student may know that a GPA of 3.5 or higher will guarantee admission to a certain college, but not the exact threshold. Similarly, a loan applicant might know that a credit score above 650 will likely suffice for securing a loan, but not the precise cutoff. These uncertainties are sometimes due to factors unknown to the agents, such as the financial situation of the lender. Therefore, agents treat the threshold as uniformly distributed within a known and reasonable range.

**Theorem D.4.** *Fix any data distribution $D$ over $\mathcal{X}$. Suppose the prior $\pi$ is uniform over an interval $[a, b]$ for some $a, b$ such that $h \in [a, b]$. Define $H_c \triangleq [c, d]$ where $d \triangleq \min(b, \max(h, f + 1))$.*

*If $b - a < 1$, we have that $H^\star = H_c$ is optimal for any $c \in [a, h]$, with corresponding utility*

$$U(H_c) = \begin{cases} 1 - \Pr_{x \sim D}[d - 1 < x < f] & d - 1 < f \\ 1 - \Pr_{x \sim D}[f \leq x \leq d - 1] & d - 1 \geq f \end{cases}$$

*If $b - a \geq 1$, we have that for any $c \in (b - 1, h]$, the optimal solution $H^\star$ is given by*

$$H^\star = \begin{cases} H_c & U(H_c) > U([a, b]) \\ [a, b] & U(H_c) \leq U([a, b]) \end{cases}$$

*where $U(H_c)$ is given above and $U([a, b]) = 1 - \Pr_{x \sim D}[f \leq x < h]$ is the utility of releasing $[a, b]$.*

*Proof of Theorem D.4.* Suppose $H = [c, d] \subseteq [a, b]$ is the released information by the learner. The agents then project their uniform prior $\pi$ over $[a, b]$ onto $H$, which leads to the uniform distribution over $[c, d]$ for $\pi|_H$, and then best respond according to $\pi|_H$. Therefore, for any agent $x$,

$$\text{BR}(x, H) = \operatorname*{argmax}_{x' \geq x} \left\{ \Pr_{h' \sim Unif[c,d]}[x' \geq h'] - (x' - x) \right\}$$

One can then show that if $d - c \geq 1$, $\text{BR}(x, H) = x$ for all $x$ because for any manipulation $(x' > x)$, the marginal gain in the probability of receiving a positive classification is less than the marginal cost.

Furthermore, if $d - c < 1$, then we have

$$\mathrm{BR}(x, H) = \begin{cases} d & d - 1 < x < d \\ x & \text{Otherwise} \end{cases}$$

Therefore, for any $H = [c, d] \subseteq [a, b]$, if $d - c \geq 1$, we have $U(H) = 1 - \Pr_{x \sim D}[f \leq x < h]$, and if $d - c < 1$, we have

$$U(H) = \begin{cases} 1 - \Pr_{x \sim D}[d - 1 < x < f] & d - 1 < f \\ 1 - \Pr_{x \sim D}[f \leq x \leq d - 1] & d - 1 \geq f \end{cases}$$

This is because under $d - c \geq 1$, no one manipulates, and thus, the error corresponds to the probability mass between $f$ and $h$: the positives who cannot manipulate to pass $h$. Under $d - c < 1$, because every agent $x > d - 1$ can receive positive classification by manipulating, the error of $H$ corresponds to the probability mass between $d - 1$ and $f$: if $d - 1 < f$, this corresponds to the negatives who can manipulate and receive positive classification, and if $d - 1 \geq f$, this corresponds to the positives who cannot manipulate to receive positive classification.

Now assume $b - a < 1$, which implies that $d - c < 1$. At a high level, to maximize $U(H)$ in this case, we want to pick $d$ such that $d - 1$ is as close as possible to $f$. More formally, our goal is to pick $[c, d] \subseteq [a, b]$ such that $h \in [c, d]$ and that the probability mass between $d - 1$ and $f$ is minimized. In this case, one can see, via a case analysis, that $d = \min(b, \max(h, f + 1))$ is the optimal value, and that $c$ can be any point in $[a, h]$.

If $b - a \geq 1$, then both $d - c < 1$ and $d - c \geq 1$ are possible. If $d - c < 1$, then the optimality of $[c, d]$ where $c$ is any point in $(b - 1, h]$, and $d = \min(b, \max(h, f + 1))$ can be established as above. Note that the choice of $c \in (b - 1, h]$ guarantees that $d - c < 1$. If $d - c \geq 1$, then the utility of the learner doesn't change if $[c, d] = [a, b]$ simply because the agents do not manipulate for any $[c, d]$ such that $d - c \geq 1$. Finally, the optimal interval is chosen based on which case ($d - c < 1$ vs. $d - c \geq 1$) leads to higher utility. $\qquad \square$

# E  The Complete Exposition of Section 4.1

In this section we will provide an efficient algorithm for computing the learner's optimal information release when the prior $\pi$ is a *discrete* uniform distribution over a set $\{h_1, h_2, \ldots, h_n\} \subseteq \mathcal{H}$ that includes the adopted classifier $h$. The objective of the learner is to release a $H \subseteq \{h_1, h_2, \ldots, h_n\}$ such that $h \in H$. Throughout, we take $h = h_k$ where $1 \leq k \leq n$, and assume $h_1 \leq h_2 \leq \ldots \leq h_n$.

We first state some facts about the agents' best response for *any* prior $\pi$ over $\{h_1, h_2, \ldots, h_n\} \subseteq \mathcal{H}$. To start, we first show that the best response of any agent can be characterized as follows:

**Lemma E.1.** *For any agent $x$, and any prior $\pi$ over $\{h_1, h_2, \ldots, h_n\} \subseteq \mathcal{H}$, we have $BR(x, H) \in \{x\} \cup \{h_i \in H : h_i > x\}$.*

*Proof of Lemma E.1.* Recall from Lemma D.2 that $\mathrm{BR}(x, H) \geq x$. Note that the utility of the agent $x \in \mathcal{X}$ from manipulating to a point $x' \geq x$ can be expressed as

$$u_x(x', H) = \sum_{i : h_i \leq x'} \pi|_H(h_i) - (x' - x)$$

For any $x' \geq x$ such $x' \notin \{x\} \cup \{h_i \in H : h_i > x\}$, it is easy to see that the agent can increase her utility by moving to a point in $\{x\} \cup \{h_i \in H : h_i > x\}$, which proves the lemma. $\qquad \square$

This lemma basically tells us that the best response of any agent $x$ is either to stay at its location, or to manipulate to $h_i \in H$ such that $h_i > x$. Given such characterization of the agents' best response in our setup, we now characterize, for any classifier $h_i$ in the support of $\pi$, the set of agents that will manipulate to $h_i$.

**Lemma E.2.** *Fix any prior $\pi$ over $\{h_1, \ldots, h_n\}$ and any $H$. If for any $i$, $\{x : BR(x, H) = h_i\} \neq \emptyset$, then for some $\alpha$, $\{x : BR(x, H) = h_i\} = (\alpha, h_i]$, where $\alpha$ satisfies $u_\alpha(\alpha, H) = u_\alpha(h_i, H)$.*

*Proof of Lemma E.2.* Let $\alpha = \inf \{x : \text{BR}(x, H) = h_i\}$. Take any $x \in (\alpha, h_i]$. We have, by the definition of $\alpha$, that there exists $x' \in (\alpha, x)$ such that $x' \in \{x : \text{BR}(x, H) = h_i\}$, implying $\text{BR}(x', H) = h_i$. Therefore, $\text{BR}(x', H) \geq x$. The third part of Lemma D.2 implies that $h_i = \text{BR}(x', H) = \text{BR}(x, H)$. This proves that

$$(\alpha, h_i] \subseteq \{x : \text{BR}(x, H) = h_i\}$$

If $x > h_i$, then $\text{BR}(x, H) > h_i$ by the first part of Lemma D.2. Therefore $x \notin \{x : \text{BR}(x, H) = h_i\}$.

If $x < \alpha$, then $\text{BR}(x, H) < h_i$ by the definition of $\alpha$, implying $x \notin \{x : \text{BR}(x, H) = h_i\}$.

If $x = \alpha$, we will show that $\text{BR}(x, H) = \alpha$. Note that $\alpha \leq \text{BR}(\alpha, H) \leq h_i$ by Lemma D.2. But if $\text{BR}(\alpha, H) > \alpha$, then by the third part of Lemma D.2, $\text{BR}(\alpha, H) = h_i$. So $\text{BR}(\alpha, H) \in \{\alpha, h_i\}$. Suppose $\text{BR}(\alpha, H) = h_i$. Therefore, there exists $\epsilon > 0$ such that $u_\alpha(\alpha, H) + \epsilon < u_\alpha(h_i, H)$, by the definition of agents' best response and the fact that tie-breaking is in favor of smaller values (Remark D.1). Let $\epsilon' \in (0, \epsilon/2]$ be such that $\{h' \in H : \alpha - \epsilon' \leq h' < \alpha\} = \emptyset$. Consider $x' = \alpha - \epsilon'$. We have, by Lemma D.2 and E.1, that $\text{BR}(x', H) \in \{x'\} \cup [\alpha, h_i]$. But because $\text{BR}(\alpha, H) = h_i$, Lemma D.2 implies that $\text{BR}(x', H) \in \{x', h_i\}$. Note that

$$u_{x'}(x', H) \leq u_\alpha(\alpha, H) < u_\alpha(h_i, H) - \epsilon = u_{x'}(h_i, H) - (\epsilon - \epsilon')$$

implying that $\text{BR}(x', H) = h_i$. But $x' = \alpha - \epsilon'$ and this contradicts with the definition of $\alpha$. Therefore $\text{BR}(\alpha, H) = \alpha$, and this completes the proof of the first part of the lemma.

We now focus on the second part of the lemma. Note that $u_\alpha(\alpha, H) \geq u_\alpha(h_i, H)$, because if $u_\alpha(\alpha, H) < u_\alpha(h_i, H)$, then $\alpha < \text{BR}(\alpha, H) \leq h_i$. Together with the first part of this lemma, and Lemma D.2, this implies that $\text{BR}(\alpha, H) = h_i$ which is a contradiction with the first part of the lemma. Next, we show that $u_\alpha(\alpha, H) \leq u_\alpha(h_i, H)$. Suppose $u_\alpha(\alpha, H) > u_\alpha(h_i, H) + \epsilon$ for some $\epsilon > 0$. Consider $x = \alpha + \epsilon/2$. We have that

$$u_x(x, H) \geq u_\alpha(\alpha, H) > u_\alpha(h_i, H) + \epsilon = u_x(h_i, H) + \epsilon/2$$

implying that $\text{BR}(x, H) \neq h_i$. This is in contradiction with the first part of the lemma. Therefore, $u_\alpha(\alpha, H) = u_\alpha(h_i, H)$. $\qquad \square$

Next, we characterize the utility of any subset $H$ released by the learner using a real-valued function of $H$. Define, for any $H \subseteq \{h_1, \ldots, h_n\}$ such that $h \in H$,

$$R_H \triangleq \inf \{x : \text{BR}(x, H) \geq h\} \tag{10}$$

Note that $\text{BR}(x = h, H) \geq h$ for any $H$ such that $h \in H$. Therefore, $\{x : \text{BR}(x, H) \geq h\}$ is nonempty, and that $R_H \leq h$ for any $H$ such that $h \in H$. Our next lemma shows that $R_H$ characterizes the utility of $H$ for the learner, for any prior $\pi$ over $\{h_1, \ldots, h_n\}$.

**Lemma 4.4.** *Fix any prior $\pi$ over $\{h_1, \ldots, h_n\}$. We have that for any $H \subseteq \{h_1, \ldots, h_n\}$ such that $h \in H$, the utility of the learner, given by Equation 2, can be written as*

$$U(H) = \begin{cases} 1 - \Pr_{x \sim D}[R_H < x < f] & R_H < f \\ 1 - \Pr_{x \sim D}[f \leq x \leq R_H] & R_H \geq f \end{cases} \tag{5}$$

*Proof of Lemma 4.4.* Recall that $U(H) = \Pr_{x \sim D}[h(\text{BR}(x, H)) = f(x)]$. The claim follows from the fact that $h(\text{BR}(x, H)) = 1$ if and only if $x > R_H$. Note that if $x > R_H$, then $\text{BR}(x, H) \geq h$ (equivalently, $h(\text{BR}(x, H)) = 1$) by the definition of $R_H$ and Lemma D.2. Further, if $\text{BR}(x, H) \geq h$ then $x > R_H$ by the definition of $R_H$. $\qquad \square$

Given such characterization of the learner's utility, we will show that when the agents' prior is uniform over $\{h_1, \ldots, h_n\}$, there are only *polynomially many* possible values that $R_H$ can take, even though *there are exponentially many $H$'s*. Our algorithm then for any possible value $R$ of $R_H$, finds a subset $H$ such that $R_H = R$, if such $H$ exists. The algorithm then outputs the $H$ with maximal utility according to Equation 5. More formally, we consider the following partitioning of the space of subsets of $\{h_1, \ldots, h_n\}$. For any $\ell \in \{1, 2, \ldots, n\}$, and for any $i \in \{k, k+1, \ldots, n\}$[13], define

$$S_{i,\ell} = \{H \subseteq \{h_1, \ldots, h_n\} : h \in H, |H| = \ell, \text{BR}(h, H) = h_i\}$$

---

[13] Recall $k$ is the index of $h$ in $\{h_1, \ldots, h_n\}$, i.e., $h = h_k$.

Note that by Lemma E.1, $\mathrm{BR}(h \equiv h_k, H) \in \{h_i : i \geq k\}$ for any $H$. Therefore, $\{S_{i,\ell}\}_{i,\ell}$ gives us a proper partitioning of the space of subsets, which implies

$$\max_{H \subseteq \{h_1,\ldots,h_n\}, h \in H} U(H) = \max_{i,\ell} \max_{H \in S_{i,\ell}} U(H)$$

We will show that when the prior is uniform, solving $\max_{H \in S_{i,\ell}} U(H)$ can be done efficiently, by showing a construction of the optimal $H \in S_{i,\ell}$ in our algorithm. To do so, we first show that $R_H$ (defined in Equation 10) can be characterized by $h_i$, when we restrict ourselves to $H \in S_{i,\ell}$.

**Lemma E.3.** *Fix any prior $\pi$ over $\{h_1,\ldots,h_n\}$. If $H \in S_{i,\ell}$, then $\{x : \mathrm{BR}(x, H) = h_i\} = (R_H, h_i]$.*

*Proof of Lemma E.3.* We first show that $R_H = \inf \{x : \mathrm{BR}(x, H) = h_i\}$. Fix any $H \in S_{i,\ell}$. Let $Q_H = \inf \{x : \mathrm{BR}(x, H) = h_i\}$. First note that because $H \in S_{i,\ell}$, we have $\mathrm{BR}(h, H) = h_i$, and therefore $\{x : \mathrm{BR}(x, H) = h_i\} \neq \emptyset$, and that $Q_H \leq h$. Additionally, because

$$\{x : \mathrm{BR}(x, H) = h_i\} \subseteq \{x : \mathrm{BR}(x, H) \geq h\}$$

we have that $Q_H \geq R_H$. So we have $R_H \leq Q_H \leq h$. If $Q_H \neq R_H$, then there exists $R_H < x < Q_H$, such that $h \leq \mathrm{BR}(x, H) < h_i$. But, for $x' = \mathrm{BR}(x, H)$, we have $\mathrm{BR}(x', H) = h_i > x' = \mathrm{BR}(x, H)$. This is in contradiction with the third part of Lemma D.2 (taking $x_1 = x$, and $x_2 = x'$). Therefore, $Q_H = R_H$, and this proves the first part of the lemma. The second part of the lemma is followed from part one and Lemma E.2. $\square$

In particular, this Lemma implies that for $H \in S_{i,\ell}$, we have $R_H = \inf \{x : \mathrm{BR}(x, H) = h_i\}$. Next, we demonstrate the possible values that $R_H$ can take for uniform priors. In particular, the following lemma establishes that $R_H$ can take only polynomially many values.

**Lemma E.4.** *If the prior $\pi$ is uniform over $\{h_1,\ldots,h_n\}$, then for any $H \in S_{i,l}$, $R_H = h_i - j/\ell$ where $j = |\{h' \in H : h' \in (R_H, h_i]\}|$.*

*Proof of Lemma E.4.* Fix any $H \in S_{i,\ell}$. Note that Lemma E.3 and Lemma E.2 together imply that $u_{R_H}(R_H, H) = u_{R_H}(h_i, H)$. This implies

$$\Pr_{h' \sim \pi|_H} [R_H \geq h'] = \Pr_{h' \sim \pi|_H} [h_i \geq h'] + (h_i - R_H)$$

But $\Pr_{h' \sim \pi|_H} [R_H \geq h'] = j_1/\ell$ and $\Pr_{h' \sim \pi|_H} [R_H \geq h'] = j_2/\ell$ where $j_1$ and $j_2$ are the number of hypotheses in $H$ that are smaller (or equal to) $R_H$, and smaller (or equal to) $h_i$, respectively. In other words,

$$j_1 = |\{h' \in H : h' \leq R_H\}|, \quad j_2 = |\{h' \in H : h' \leq h_i\}|$$

Therefore,

$$R_H = h_i - \frac{j_2 - j_1}{\ell}$$

which completes the proof. $\square$

Given such characterization, Algorithm 3, for any $i, \ell$, enumerates over all possible $j \in \{1, \ldots, \ell\}$ and returns a $H$ such that $H \in S_{i,\ell}$ and $R_H = h_i - j/\ell$, if such $H$ exists. To elaborate, for any $i, \ell, j$, such $H \equiv H_j^{i,\ell}$ is constructed by first picking the $j$ largest classifiers that are between $h_i - j/\ell$ and $h_i$ (including both $h_i$ and $h$). If there are not at least $j$ classifiers between $h_i - j/\ell$ and $h_i$, then no such $H$ exists for $i, \ell, j$ because of Lemma E.4. After picking the first $j$ elements as described, the remaining $\ell - j$ classifiers are first chosen from all classifiers that are less than (or equal to) $h_i - j/\ell$, and once these classifiers are exhausted, the rest are taken from the classifiers that are larger than $h_i$, starting from the largest possible classifier, and going down until $\ell$ classifiers are picked.

Note that this construction of $H \equiv H_j^{i,\ell}$ guarantees that $\mathrm{BR}(h_i, H')$ is minimized among all $H'$'s with corresponding values of $(i, \ell, j)$. Therefore, if $\mathrm{BR}(h_i, H) > h_i$, it is guaranteed that no $H$ exists for $(i, \ell, j)$. If $\mathrm{BR}(h_i, H) = h_i$, then the construction of $H \equiv H_j^{i,\ell}$ guarantees that $R_H = \inf \{x : \mathrm{BR}(x, H) = h_i\}$ is as small as possible. Therefore, if $\inf \{x : \mathrm{BR}(x, H) = h_i\} > h_i - j/\ell$, then it is guaranteed that no such $H$ exists for $(i, \ell, j)$ (note that $\inf \{x : \mathrm{BR}(x, H) = h_i\} \geq h_i - j/\ell$ by construction). The algorithm finally outputs, among all $H$'s found, the subset $H$ with maximum utility according to Equation 5.

This proves the following theorem.

---

**Algorithm 3:** The Learner's Optimization Problem: Discrete Uniform Prior

---

**Input:** ground truth classifier $f$, adopted classifier $h \geq f$, prior's support $\{h_1, \ldots, h_n\}$ where
      $h_1 \leq \ldots \leq h_n$ and $h_k = h$, data distribution $D$

**for** $i = k, k+1, \ldots, n$ **do**
    **for** $\ell = 1, 2, \ldots, n$ **do**
        **for** $j = 1, 2, \ldots, \ell$ **do**
            $R \leftarrow h_i - j/\ell$ ;                     `// candidate value R for` $R_H$`.`
            $S_1 \leftarrow \{h' \in \{h_1, \ldots, h_n\} : R < h' \leq h_i\}$ ;    `// all classifiers between R and`
            $h_i$. 
            $S_2 \leftarrow \{h' \in \{h_1, \ldots, h_n\} : h' \leq R\}$ ;        `// all classifiers smaller than R.`
            $S_3 \leftarrow \{h' \in \{h_1, \ldots, h_n\} : h' > h_i\}$ ;         `// all classifiers larger than` $h_i$`.`
            **if** $R \geq h$ *or* $|S_1| < j$ **then**
                $H_j^{i,\ell} \leftarrow \perp$ ;                        `// no H exists for` $(i, \ell, j)$`.`
            **else**
                $H_j^{i,\ell} \leftarrow \{h, h_i\}$;
                $H_j^{i,\ell} \leftarrow H_j^{i,\ell} \cup \mathrm{MAX}_{j - |H_j^{i,\ell}|} \left( S_1 \setminus H_j^{i,\ell} \right)$ ;    `// MAX`$_m(\cdot) \triangleq m$ `largest elements`
                **if** $|S_2| \geq \ell - j$ **then**
                    $T \leftarrow$ any subset of size $\ell - j$ from $S_2$
                **else**
                    $T \leftarrow S_2 \cup \mathrm{MAX}_{\ell - j - |S_2|} (S_3)$ ;          `// MAX`$_m(\cdot) \triangleq m$ `largest elements`
                $H_j^{i,\ell} \leftarrow H_j^{i,\ell} \cup T$;
                **if** $BR(h_i, H_j^{i,\ell}) > h_i$ **then**
                    $H_j^{i,\ell} \leftarrow \perp$ ;                      `// no H exists for` $(i, \ell, j)$`.`
                **else**
                  **if** $\inf \left\{ x : BR(x, H_j^{i,\ell}) = h_i \right\} > R$ **then**
                      $H_j^{i,\ell} \leftarrow \perp$ ;                 `// no H exists for` $(i, \ell, j)$`.`
            **if** $H_j^{i,\ell} = \perp$ **then**
                $U_j^{i,\ell} \leftarrow -\infty$;
            **else**
                **if** $R < f$ **then**
                    $U_j^{i,\ell} \leftarrow 1 - \Pr_{x \sim D}[R < x < f]$ ;    `// utility according to Equation 5.`
                **if** $R \geq f$ **then**
                    $U_j^{i,\ell} \leftarrow 1 - \Pr_{x \sim D}[f \leq x \leq R]$ ;    `// utility according to Equation 5.`

**Output:** $H^\star = H_{j^\star}^{i^\star, \ell^\star}$ where $(i^\star, \ell^\star, j^\star) \in \mathrm{argmax}_{(i,\ell,j)} U_j^{i,\ell}$.

---

**Theorem 4.6.** *There exists an algorithm (Algorithm 3) that for any $n$, any uniform prior over $\{h_1, \ldots, h_n\}$ that includes $h$, and any data distribution $D$, returns $H^\star \subseteq \{h_1, \ldots, h_n\}$ in time $O(n^3)$ such that $h \in H^\star$, and that $U(H^\star) = \max_{H \subseteq \{h_1, \ldots, h_n\}, h \in H} U(H)$.*

## F   Missing Proof of Section 4.2

**Theorem 4.7.** *Fix any $h \geq f$. For any prior $\pi$ over $\{h_1, \ldots, h_n\}$ that includes $h$, we have 1) $FPR(\mathcal{H}) \leq FPR(\{h\})$. 2) $FNR(\{h\}) \leq FNR(H)$ for every $H \subseteq \mathcal{H}$ such that $h \in H$.*

*Proof of Theorem 4.7.* We begin by showing that $FPR(\mathcal{H}) \leq FPR(\{h\})$. Let $x \in \mathcal{X}$ be such that $f(x) = 0$ and $h(BR(x, \mathcal{H})) = 1$. We will show that $h(BR(x, \{h\})) = 1$.

Lemma E.1 together with $h \geq f$ imply the existence of $h_j$ such that $BR(x, \mathcal{H}) = h_j > x$ (as $f(x) \neq h(BR(x, \mathcal{H}))$). This further indicates that when $\mathcal{H}$ is released, the utility of the agent is strictly higher when it manipulates to $h_j$, compared to not moving:

$$u_x(h_j, \mathcal{H}) = \sum_{i=1}^{j} \pi(h_i) - (h_j - x) > \sum_{i : h_i \leq x} \pi(h_i) = u_x(x, \mathcal{H})$$

Note that $h(BR(x, \mathcal{H})) = 1$ and $BR(x, \mathcal{H}) = h_j$ implies that $h_j \geq h$, and therefore:

$$u_x(h, \{h\}) = 1 - (h - x) \geq \sum_{i=1}^{j} \pi(h_i) - (h_j - x) > \sum_{i:h_i \leq x} \pi(h_k) = u_x(x, \{h\})$$

Since Lemma E.1 implies that $BR(x, \{h\}) \in \{x, h\}$, we derive from the above inequality that $h(BR(x, \{h\})) = 1$. This proves the first part of the theorem.

Next, we show that $FNR(\{h\}) \leq FNR(H)$ for every $H \subseteq \mathcal{H}$. Let $x \in \mathcal{X}$ be such that $f(x) = 1$ and $h(BR(x, \{h\})) = 0$, and let $H$ be any subset of $\mathcal{H}$. We will show that $h(BR(x, H)) = 0$.

Lemma E.1 implies that $BR(x, \{h\}) \in \{x, h\}$. Together with $h(BR(x, \{h\})) = 0$, we derive that $BR(x, \{h\}) = x$, and thus:

$$u_x(h, \{h\}) = 1 - (h - x) \leq u_x(x, \{h\}) = 0$$

Now, for every $h_j$ such that $h_j \geq h$, we have:

$$u_x(h_j, H) = \sum_{i=1}^{j} \pi|_H(h_i) - (h_j - x) \leq 1 - (h - x) \leq 0 \leq \sum_{i:h_i \leq x} \pi|_H(h_i) = u_x(x, H).$$

As a result, when the learner releases $H$, the utility of agent $x$ from remaining at $x$ is greater than (or equal to) any manipulation $h_j$ such that $h_j \geq h$. This implies that $h(BR(x, H)) = 0$. $\square$

## G   Optimal Information Release for Minimizing FPR

We show that minimizing FPR, unlike minimizing FNR, does not always have a clear optimal solution for the learner, by providing three examples with very different optimal solutions.

**Example G.1** (Full-information-release is optimal for FPR). Fix any $B > 1$ and any $0 \leq t < B - 1$. Let $D$ be the uniform distribution over $\mathcal{X} = [0, B]$, and $f(x) = \mathbb{1}[x \geq t]$. Let $\mathcal{H}$ be the class of single-sided threshold classifiers and suppose the adopted classifier $h(x) = \mathbb{1}[x \geq t + 1]$. Under any prior over $\mathcal{H}$, one can show that the full information release of $H = \{h\}$ achieves perfect FPR for this setting: $FPR(\{h\}) = 0$.

**Example G.2** (No-information-release is optimal for FPR). Under the same setup as in Example 2.4, one can show that releasing the support of the prior $H = \{h_{1.8}, h_2\}$ achieves $FPR(H) = 0$, whereas full information release of the adopted classifier $h = h_2$ achieves $FPR(\{h\}) = 0.9/1.9 \approx 0.47$. Note that $H = \{h_{1.8}, h_2\}$ is the support of the prior, so it constitutes as no-information-release. In other words, we have $FPR(H') = FPR(H) = 0$ for every $H'$ such that $H \subseteq H' \subseteq \mathcal{H}$.

**Claim G.3.** *There exists an instance in which neither full-information-release nor no-information-release are optimal when the utility function of the learner is $U_{FPR}$.*[14]

*Proof of Claim G.3.* We construct such an instance as follows. Suppose the domain is $\mathcal{X} = \{x_1, x_2\}$ with $x_1 = 0$, $x_2 = 0.4$ and the distribution $D$ is given by $D(x_1) = D(x_2) = 0.5$. In addition, consider $f = 0.3$, and hypothesis class $\mathcal{H} = \{h_1, h_2, h_3\}$ where $h_1 = 0.1, h_2 = 0.5, h_3 = 0.7$, and a prior distribution $\pi$ such that $\pi(h_1) = 0.2, \pi(h_2) = 0.1, \pi(h_3) = 0.7$.

Observe that under full-information-release, $FPR(\{h\}) = 1$ for every $h \in \mathcal{H}$. Now suppose $h = h_2$ is the adopted classifier. We have that $BR(x_1, \{h\}) = 0.5$ implying $h(BR(x_1, \{h\})) = 1 \neq f(x_1) = 0$ implying $x_1$ is a false positive under $\{h\}$ release. Additionally, $BR(x_1, \mathcal{H}) = 0.7$ implies that $h(BR(x_1, \mathcal{H})) = 1 \neq f(x_1) = 0$ implying $x_1$ is a false positive under $\mathcal{H}$ release. Further, it holds that $BR(x_1, \{h_1, h_2\}) = 0.1$ and so $h(BR(x_1, \{h_1, h_2\})) = 0 = f(x_1)$. Moreover, in this particular instance, releasing $\{h_1, h_2\}$ achieves perfect utility as $BR(x_2, \{h_1, h_2\}) = 0.5$ which implies $h(BR(x_2, \{h_1, h_2\})) = 1 = f(x_2)$. $\square$

## H   Possible Information Leakage Through Firm's Choice of $H$

One limitation of our model is that if the agents have knowledge of the mapping $f$ from feature vectors to labels, they might gain information on $h^*$ in cases when $|H| > 1$. More specifically,

---

[14]We remark that the claim holds when the utility function is $U$ (as defined in Equation 2) as well.

knowing the mapping $D$, the mapping $f$, and that the firm is optimizing the choice of $H$ for accuracy, agents could deduce $h^*$. In this case, the choice of $H$ leaks more information than intended.

We proceed by showing an example with threshold classifiers for such a case.

**Example H.1.** Suppose a distribution $D$ over agents $x \in \mathcal{X} = [0,1]$ is uniform, $c(x, x') = |x - x'|$, and $f(x) = \mathbb{1}[x \geq 0.15]$. The set of classifiers available to the firm is $\mathcal{H} = \{h_1, h_2\}$ where $\alpha_1 = 0.1, \alpha_2 = 0.9$ is each classifier's respective threshold.

Following our Bayesian model, the firm chooses to release a subset $H \subseteq \mathcal{H}$ over which the agents have a uniform prior $\pi|_H$. The agents know that the firm is choosing $H$ to optimize accuracy, i.e. the function $U(H) = \Pr_{x \sim D}[h^*(\Delta_H(x)) = y]$ (where $y = \mathbb{1}[x \geq T]$). We will show that the firm can release a subset $H$ which is in $\arg\max_{H \subseteq \mathcal{H}}[U(H)|h^* = h_1]$ but *not* in $\arg\max_{H \subseteq \mathcal{H}}[U(H)|h^* = h_2]$, allowing the agent to reason that $h^*$ must be $h_1$.

**Proposition H.2.** *Consider Example H.1. If the agents know agents know that $f(x) = \mathbb{1}[x \geq 0.15]$ and the prior $D$, they can infer that $h^* = h_1$.*

*Proof.* We first solve for $\arg\max_{H \subseteq \mathcal{H}}[U(H)|h^* = h_1]$. Suppose $H = \{h_1\}$. Then all agents know $h^* = h_1$. If $x \leq \alpha_1$, the agent will manipulate to $\alpha_1$ to get a positive outcome if $1 - c(x, \alpha_1) > 0$, which is always true. If $x \geq \alpha_1$, agents will stay the same to get a positive outcome. So the set of misclassified agents is those with $x \in [0, T]$ and $U(\{h_1\}) = 1 - T = 0.85$. Now suppose $H = \{h_1, h_2\}$. Agents believe each classifier is $h^*$ with probability 0.5. If $x \geq \alpha_2$, agents are guaranteed a positive outcome and stay the same. If $\alpha_1 \leq x \leq \alpha_2$, agents will manipulate to $\alpha_2$ to get a positive outcome if $1 - c(x, \alpha_2) > 0.5$, so all agents with $x \in (0.4, 0.9]$ will be classified correctly. The rest of the agents with $x \in [\alpha_1, 0.4]$ will stay the same to get a positive outcome with probability 0.5, and those with $x \in [\alpha_1, T]$ will be misclassified. Lastly, if $x < \alpha_1$, agents will manipulate to $\alpha_2$ to get a guaranteed positive outcome if $1 - c(x, \alpha_2) > 0.5 - c(x, \alpha_1)$ (a.k.a. $1 - 0.9 - x > 0.5 - 0.1 - x$), which is never true, and otherwise manipulate to $\alpha_1$ to get a positive outcome with probability 0.5 if $0.5 - c(x, \alpha_1) > 0$, which is always true. So the set of misclassified agents is those with $x \in [0, T]$ and $U(\{h_1, h_2\}) = 1 - T = 0.85$. Therefore $\arg\max_{H \subseteq \mathcal{H}}[U(H)|h^* = h_1] = \{\{h_1\}, \{h_1, h_2\}\}$.

Now we consider $\arg\max_{H \subseteq \mathcal{H}}[U(H)|h^* = h_2]$. Suppose $H = \{h_2\}$. As before, all agents know $h^* = h_2$. If $x \leq \alpha_2$, the agent will manipulate to $\alpha_2$ to get a positive outcome if $1 - c(x, \alpha_2) > 0$, which is always true. If $x \geq \alpha_2$, agents will stay the same to get a positive outcome. So the set of misclassified agents is those with $x \in [0, T]$ and $U(\{h_2\}) = 1 - T = 0.85$. Now suppose $H = \{h_1, h_2\}$. As before, agents believe each classifier is $h^*$ with probability 0.5. If $x \geq \alpha_2$, agents are guaranteed a positive outcome and stay the same. If $\alpha_1 \leq x \leq \alpha_2$, agents will manipulate to $\alpha_2$ to get a positive outcome if $1 - c(x, \alpha_2) > 0.5$, so all agents with $x \in (0.4, 0.9]$ will be classified correctly. The rest of the agents with $x \in [\alpha_1, 0.4]$ will stay the same to get a positive outcome with probability 0.5, and will be misclassified. Lastly, if $x < \alpha_1$, agents will manipulate to $\alpha_1$ to get a positive outcome with probability 0.5, and will be classified correctly. So the set of misclassified agents is those with $x \in (0.4, 0.9]$ and $U(\{h_1, h_2\}) = 1 - |0.9 - 0.4| = 0.5$, which is less than for $U(\{h_2\})$. Therefore $\arg\max_{H \subseteq \mathcal{H}}[U(H)|h^* = h_2] = \{\{h_2\}\}$.

We have shown that if the firm chooses $H^* = \{h_1, h_2\}$, the agent can reason that $H^* \in \arg\max_{H \subseteq \mathcal{H}}[U(H)|h^* = h_1]$ but $H^* \notin \arg\max_{H \subseteq \mathcal{H}}[U(H)|h^* = h_2]$, so $h^*$ must be $h_1$. $\qquad\square$

