# OpenReview forum: "Bayesian Strategic Classification"
_NeurIPS.cc/2024/Conference — NeurIPS 2024 poster_

### Official Review · Reviewer_VHeD · 2024-07-10

**Soundness:** 4
**Presentation:** 3
**Contribution:** 2
**Rating:** 5
**Confidence:** 3

**Summary:**

The paper studies partial information release in strategic classification.  Roughly speaking, the learner publishes a set of classifiers containing the actual one being used, and the agents update their beliefs accordingly and best respond.  The authors show that the agents' problem of best responding is generally hard, and give oracle-efficient (approximate) algorithms in certain cases.  They then consider the essentially one-dimensional case where quality is measured by a single number, and show (1) the problem of releasing the optimal set is generally hard, and (2) an efficient algorithm for uniform priors.

**Strengths:**

Conceptually, the problem of "ambiguous" classifiers is natural and interesting.  The paper makes an attempt at modeling the problem and presents a variety of results, which might trigger further progress.  The paper (especially the introductory parts) is also quite well written and polished.

**Weaknesses:**

My two major concerns are regarding the model and the significance of section 4.  I feel certain parts of the model can be better justified.  I also feel that in some sense section 4 is trying to solve a problem that might not exist in the first place.  See detailed comments below.

**Questions:**

The model: while the model in the paper makes sense, I'm also curious about other natural ways to model the problem, e.g., the learner could commit to a (possibly restricted) distribution over classifiers and hide the realization from the agent(s).  Is there anything interesting we can say about such alternative models?  Is there a reason to focus on the particular model in this paper?  In particular, I guess one potential criticism of the current model is that, given enough time, agents will eventually figure out the actual classifier being used (e.g., in the same way that they come up with the prior distribution).

Line 168, "truthfully": I'm a bit unsure about the wording here.  In particular, the posterior of the agents is generally misleading.  More generally, there seems to be a mismatch between what the agents know about the learner and how they behave -- the agents have no reason to believe that the classifier being used is distributed according to the posterior.  I wonder what the authors think about this.

Def 2.2: this will probably become clear soon, but is h part of the learner's action?  It sounds like it's not, but then why do you say this is a generalization of the full information release game?

Line 314, h \ge f: notation abused here.

Sec 4: in this setup, why shouldn't the learner simply publish the exact optimal classifier (h = f + 1, which is perfectly accurate after manipulation)?  I guess this is also where fixing h as an input parameter makes less sense.

**Limitations:**

I don't have concerns.

---

> ### Author Rebuttal · Authors · 2024-08-07
>
> Thank you for these questions, we will now respond in detail.
>
> **The model (while the model in the paper makes sense, I'm also curious about other natural ways to model the problem, e.g., the learner could commit to a (possibly restricted) distribution over classifiers and hide the realization from the agent(s). Is there anything interesting we can say about such alternative models? Is there a reason to focus on the particular model in this paper? In particular, I guess one potential criticism of the current model is that, given enough time, agents will eventually figure out the actual classifier being used (e.g., in the same way that they come up with the prior distribution).):**
>
> While working with randomized classifiers is an interesting direction for future work, it is still susceptible to the same criticism of the reviewer: over time the agents will be able to learn what distribution the learner is using and best respond according to that. Also, in practice, it may be undesirable for a learner to constantly change their deployed classifier, both because it is costly but also because it may be perceived as unreliable or unfair.
>
> We believe our model is a natural and simple first attempt towards understanding incomplete information in strategic classification. Having said that, we completely agree with the reviewer that the use of randomized models is an interesting direction for future work, as we have mentioned it in the limitations section: “... However, we note that changing the screening algorithm requires significant resources, and the rate at which the classifier is updated is generally slower than the rate at which decisions are made. In practice, this means that strategic agents effectively face a fixed model in each “batch” between updates.”
>
> **Line 168, “truthfully” (I'm a bit unsure about the wording here. In particular, the posterior of the agents is generally misleading. More generally, there seems to be a mismatch between what the agents know about the learner and how they behave -- the agents have no reason to believe that the classifier being used is distributed according to the posterior. I wonder what the authors think about this.):**
>
>
>
> We call it truthful because we require that the learner includes its deployed classifier $h$ in the shortlist $H$. In other words, we don’t allow the learner to `lie’ to the agents. Moreover, after releasing $H$, regardless of what the common prior distribution $\pi$ is, the probability of $h$ in the posterior $\pi \vert_H$ can only increase.
>
> We note that the behavior of our agents follow a traditional Bayes update, the natural model used for updating informational beliefs in the vast majority of related game theory and in particular Bayesian Persuasion [3] literature. Truthfulness and believing that the learner is truthful are also the standard assumption in settings involving information revelation [3], with the motivation that these games are played in a repeated context where lying is observable and punishable in the long-term.
>
>
>
> **Def 2.2 (this will probably become clear soon, but is h part of the learner's action? It sounds like it's not, but then why do you say this is a generalization of the full information release game?):**
>
> The reviewer is correct that if $h$ is not the strategic optimal classifier (i.e., one that maximizes the strategic accuracy in the full information release game), this will not be a generalization of the full information release game. We will clarify it in the final version of the paper.
>
> We assume that $h$ is fixed and the learner is not willing to change it due to cost issues: as mentioned in the paper, “... However, we note that changing the screening algorithm requires significant resources, and the rate at which the classifier is updated is generally slower than the rate at which decisions are made”.
>
> We further note that in the case where the fixed $h$ is the strategic optimal classifier in the full information release game (i.e. “standard” strategic classification), our game essentially becomes a generalization of the full information release game because the learner can always choose $H=\{h\}$.
>
> **Line 314, $h \ge f$: notation abused here.** Remark 4.2 explains the notational abuse that we adopt for simplicity. We simply view $f$ (or $h$) both as a function and their corresponding real-valued threshold, so $h \ge f$ simply means that the threshold for $h$ is larger than or equal to the threshold for $f$.
>
> **Sec 4: why not publish $h = f + 1$ (in this setup, why shouldn't the learner simply publish the exact optimal classifier ($h = f + 1$, which is perfectly accurate after manipulation)? I guess this is also where fixing h as an input parameter makes less sense.):**
>
> Releasing $h=f+1$ is not necessarily an optimal or even a feasible solution because in general the space of thresholds is bounded (which is natural because test scores are normally bounded, and so must be the thresholds we work with on those test scores). Observe that this is shown in Example 2.4.
>
> [3] Kamenica, Emir, and Matthew Gentzkow. "Bayesian persuasion." American Economic Review 101, no. 6 (2011): 2590-2615.

---

> > ### Comment · Reviewer_VHeD · 2024-08-08
> >
> > Thank you for your response!  It answers most of my questions (which are somewhat open-ended in the first place).  I agree that the model is reasonable given that the paper is an early attempt to study the phenomenon.  Accordingly, I will increase my score to 5.

---

### Official Review · Reviewer_XrSi · 2024-07-12

**Soundness:** 3
**Presentation:** 3
**Contribution:** 3
**Rating:** 5
**Confidence:** 3

**Summary:**

The paper studies strategic classification problems where agents with partial information about the classifier can strategically report themselves at a cost. The problem is modeled as a Stackelberg game: The principal can first reveal partial information of the classifier, then the agents choose their strategy to report. The goal of the principal is to maximize the accuracy of the classification as if they know the true value of all the agents. They show that this partial information release from the principal, compared to releasing no information or all information, may improve the accuracy. The theoretical results include NP-hardness and efficient algorithms.

**Strengths:**

+ The problem of strategic classifying is well-motivated. The characterization of the Bayesian setting and partial information release is very reasonable.
+ The Stackelberg model of the problem is reasonable.
+ The technical results are non-trivial.

**Weaknesses:**

- All the positive results seem to be under strong constraints. This may imply that the problem itself is very difficult.
- I feel that the title does not reflect the main idea of this paper. What I get from the introduction is that the key point is "partial information release" rather than "Bayesian", but partial information release is not mentioned at all in the title. I would prefer something like "Partial Information Release in Bayesian Strategic Classification".
- It's kind of unclear what there is an oracle in the agent's best response problem and what does the oracle do.

**Questions:**

1. Could you describe more clearly and intuitively what the oracle does? Also why it requires an input as $R^+ \cap R^-$?
2. Why do you take an oracle into consideration in the agent best-response problem?
3. Remark 4.3: I don't quite get this part. Why the learner what to make the classification "harder"? Can you propose an example?

**Limitations:**

Yes.

---

> ### Author Rebuttal · Authors · 2024-08-07
>
> Thank you for this review, we will now respond in detail.
>
> **Positive results seem to be under strong constraints. This may imply that the problem itself is very difficult:**
>
> A challenging aspect of work in strategic classification theory, along with other fields in ML is that with minimal assumptions, many problems are intractable.
>
> For instance, we show that the learner’s optimal information release problem is NP-hard when the agents’ prior can be arbitrary, and a similar result is exhibited in the first strategic classification paper [Hardt et al. 2016].
>
> That said, despite such hardness barriers, adding some assumptions to relax the problem is important so we can investigate specific challenges, craft initial algorithms that then can be used as a foundation for later empirically performant algorithms.
>
> **Role of the oracle:**
> Description of the oracle: the oracle solves the following “projection” problem: given an agent $x$ with cost function $c$, a collection of classifiers $\{h_1, \ldots, h_n\}$, and a binary (0/1) vector $b=(b_1, \ldots, b_n)$, the oracle returns a point $z$ that minimizes the cost $c(x,z)$ for the agent such that for all $i$, $h_i (z) = b_i$. Instead of using the binary vector $b$, in our paper we adopt $R^+$ (classifiers whose corresponding $b_i$ is $1$) and $R^-$ (classifiers whose corresponding $b_i$ is $0$). Note that when $n=1$, this is simply projecting the point $x$ onto either $\{h_1(z) = 1\}$ or $\{h_1(z) = 0\}$, according to the cost function $c$. When $n>1$, this is the projection of $x$, according to the cost function $c$, onto one of the $2^n$ regions created by the $n$ classifiers (each classifier has a positive and a negative region) which we denote by $R^+ \cap R^-$. Informally, the oracle abstracts out the task of finding the closest point, w.r.t. a given cost function, to a given point (an agent), that passes only the classifiers in $R^+$ without passing the classifiers in $R^-$ (or equivalently a point in $R^+ \cap R^-$ because we require to pass exactly ones corresponding to $R^+$ and nothing more)..
>
>
> **Why we use an oracle:** we need an oracle model because solving the best response problem in Equation 1 may span a large class of potential deployed classifiers, prior distribution $\pi$, and cost function $c$, and specialized optimization techniques are required for each problem class. In our setting, we want to focus on the details of information release so by assuming the oracle model we can isolate the impact of the information release, while abstracting away the agent’s ability to solve the best response problem which is tangential to our point. This allows our assumptions to model a range of agents using different optimization methods to solve Equation 1; further, we can plug in a large class of optimization methods in place of the oracle as needed for specific applications.
>
>
> **Re Question about Remark 4.3:**
> By “harder”, we mean that the classifier $h$ is more stringent and harder to pass, in that it uses a higher threshold than the true classifier $f$. The reason we need to do so is because the agents’ ability to manipulate their features allows them to obtain a *higher* perceived score than their true score. The learner needs to make the classifier more difficult to pass to compensate for this strategic behavior and inflated scores. This is relatively standard in strategic classification, where the classifier is “pushed to the right” to be robust to manipulation.

---

> > ### Comment · Reviewer_XrSi · 2024-08-08
> >
> > Thank you for your response! This answers my questions about the oracle and Remark 4.3. I have an additional question about the oracle. How should I expect the complexity of such an oracle in practical applications? That is if a strategic agent cannot optimize even on class efficiently, then calculating the overall best response would be even harder.

---

> > > ### Author Response · Authors · 2024-08-08
> > > **Response to Question**
> > >
> > > Thank you for your continued engagement in the discussion!
> > >
> > > We note that in the case where the cost function $c(x,.)$ is convex in its second argument and the agent $x$ is facing linear threshold classifiers, the optimization problem of the oracle becomes a convex program (finding the closest point to intersection of halfspaces w.r.t. a convex cost $c$) and can be solved with convex optimization techniques.
> > >
> > > In other cases, however, this problem is non-convex and can be hard in general. Having said that, the literature on non-convex optimization includes a plethora of practical optimization strategies that can handle non-convex problems. We emphasize that our major contribution is introducing and analyzing partial information release in strategic classification, and therefore, assuming such an oracle allows us to abstract away the challenges faced in non-convex optimization which is not the focus of our paper.

---

> > > > ### Comment · Reviewer_XrSi · 2024-08-12
> > > >
> > > > Thank you for your response. This makes sense to me.

---

### Official Review · Reviewer_oMrT · 2024-07-12

**Soundness:** 3
**Presentation:** 3
**Contribution:** 3
**Rating:** 6
**Confidence:** 4

**Summary:**

* This paper investigates strategic classification in a partial-information setting using a Bayesian common-prior framework.
* The setting extends standard strategic classification (Hardt et al. 2016) to a partial-information setting by assuming that the deployed classifier $h\\in\\mathcal{H}$ is not fully known to the agents, but rather assumed to be sampled from a common prior $\\pi$ over the hypothesis class. The learner has the ability to influence agent decisions by disclosing a subset of possible classifiers $H\\in\\mathcal{H}$ such that $h\\in H$. The agent moves their feature vector to maximize their expected utility (expected prediction according to posterior, minus cost). The goal of the learner is to deploy a classifier which maximizes accuracy under the agent’s best response.
* In Section 3, the authors consider the agents’ best-response problem. The first result shows that computing the agent’s best response requires (in the general case) exponentially-many calls to a projection oracle (where the hypothesis class is assumed to be finite and complexity is measured in terms of $n=|\\mathcal{H}|$). The second result shows an $O(n^d)$ upper bound for computing the best response when $\\mathcal{H}$ contains $n$ linear classifiers over $\\mathbb{R}^d$. In Section 4.1, an efficient algorithm is presented for optimizing the disclosure set of hypotheses $H$ over one-dimensional realizable threshold classifiers. Finally, Section 4.2 shows that no disclosure is optimal when optimizing false-positive rate, and that full disclosure is optimal when optimizing false-negative rate.

**Strengths:**

* Paper is very well-written, and easy to follow.
* Model is clean and concise, and captures an interesting aspect of information disclosure.
* Theoretical analysis presents both lower and upper bounds, helping establish a hardness spectrum that can motivate future work.

**Weaknesses:**

* The interaction model presented in the paper makes strong assumptions which may not be applicable in practical scenarios. In particular:
  * Assuming that the true relation between features and labels is known to the learner. In particular, the paper does not seem to discuss implications of learning from samples.
  * Assuming that the agents’ prior distribution $\\pi$ is known to the learner.
  * Assuming that $\\pi$ has finite support, and implying that all classifiers in the support can be enumerated in reasonable time. Common hypothesis classes (such as linear models and neural networks) are defined by a continuous parameter space, and have infinite cardinality.
* Interaction model seems to be vulnerable to learner dishonesty.
* The paper does not contain empirical evaluation, and it's not clear whether the proposed scheme is feasible in practical scenarios.
* Positive results on the learner’s optimal information disclosure are limited to one-dimensional classifiers.

**Questions:**

* Common prior in practical settings:
  * What could be reasonable assumptions for the functional form of the common prior $\\pi$ in practical settings?
  * How can $\\pi$ be estimated from data?
  * In which plausible practical scenarios does the size of $\pi$'s support induce a practically-significant discrepancy between efficient and inefficient algorithms?
* Dishonest information disclosure:
  * How much power can the learner gain from acting in a non-truthful way?
  * Can truthful information sharing (by the learner) can be enforced or incentivized?
* What are the consequences of the learner not knowing the feature-label distribution $D$, and being required to estimate it from data?
* Can the analysis framework be used to estimate the typical cardinality/"complexity" of the optimal hypothesis sets $H$ reported by the learner? Is there intuition for practical scenarios where a non-trivial $H$ is expected to be "simple to describe" or particularly complex in some sense?

**Limitations:**

Limitations are discussed in the appendix.

---

> ### Author Rebuttal · Authors · 2024-08-07
>
> Thank you for these detailed comments which we shall now do our best to answer!
>
> Part 1
>
> **Interaction model, true relation between features and labels is known to the learner:**
>
> This is an assumption made in prior work on strategic classification (e.g. see [2,4]). That being said, extending our results to a setting in which the learner only has access to labeled examples is an interesting future work.
>
> **Agents’ prior distribution is known to the learner:**
> This is a common assumption in Bayesian Persuasion and in game-theoretic Bayesian settings, e.g., [1].
>
> From a motivational point of view, for example, employers also have access to job interview questions dataset such as Glassdoor when hiring, and may have a good understanding of the information available to the agents. This prior can also encode agents having a very simple prior showing understanding top few features that are commonly known, which is reasonable in settings such as credit scoring (where everyone knows most agents understand that paying on time has a positive impact, for example). There are also natural cases where we can assume that the prior is uniform, corresponding to the agents having no prior information revelation in other Bayesian settings (e.g., active learning [5], the last section on  pg. 2.).
>
>
> **Assuming that $\pi$ has finite support:**
>
>  We will include this extension in the final version of the paper: note that one can often discretize the space of possible classifiers to reduce to a finite setting. We do note that only our results in Section 3 rely on finiteness, but small cardinality is not required, so discretization is sufficient for our purposes. We have complemented our results in Section 4 which primarily focuses on discrete uniform priors by considering the case of continuous uniform priors (which have infinitely many elements in their support); see Section D.2 in the Appendix for details.
>
> Regarding the discretization approach for Section 3, note that if $\pi$ has infinite support size, we can ignore classifiers with sufficiently small probabilities (i.e., $poly(\epsilon)$), as they do not affect the manipulation strategy when searching for a $(1+\epsilon)$-approximate solution. The number of classifiers in the support with probability at least $poly(\epsilon)$ for a fixed $\epsilon>0$ is at most $1/poly(\epsilon)$ which is a finite number. Therefore, to obtain a nearly optimal solution, it suffices to only consider probability distributions $\pi$ with finite support size.
>
>
> Also, observe that finite $\pi$ captures broad phenomena already, like job interview questions on sites like Glassdoor/Leetcode.
>
> **Common prior in practical setting (1. What could be reasonable assumptions for the functional form of the common prior $\pi$ in practical settings? 2. How can $\pi$ be estimated from data? 3. In which plausible practical scenarios does the size of $\pi$'s support induce a practically-significant discrepancy between efficient and inefficient algorithms?):**
>
> 1. In Section 4.1, we think of the prior as uniform over a set; this can be seen as initially having no information about the classifier, thinking all classifiers are equally likely on the support (the support itself could encode information about what reasonable classifiers are). This can be easily estimated from data. If there is any prior knowledge (e.g., 0.1 fraction of technical interview questions involve dynamic programming questions), we assume the prior incorporates this knowledge.
>
> 2. Estimating $\pi$ from data is a good question. In some of the more specific settings that exist in practice, like humans adapting to high stakes test/interview questions, humans do maintain tables of counts over the discrete range of possible questions, and then create the prior by dividing the count for each question by the normalization term, e.g. summing over all counts. E.g. when people take a test they then share this information with their peers who also have access to the database.
>
> 3. It is an interesting question thinking about the impact of $\pi$ on efficient/inefficient algorithms in practice. This question will require further work.
>
> **Dishonest information disclosure How much power can the learner gain from acting in a non-truthful way?**
>
> We restrict to truthful information disclosure as in Bayesian Persuasion [3]. Relaxing this requirement obviously extends the action space of the learner and hence can only increase the learner’s utility. Here is an example in which lying to the agents can achieve optimal utility for the learner: consider a case in which $X = [0,1]$, $D$ is uniform over $X$, and $f(x)=1[x \geq 0.5]$ is the ground truth and $h=f$ is the deployed classifier. If the learner lies to the agents and tells them that the deployed classifier is $h’(x)=1$ (i.e. everyone is qualified), regardless of what the prior distribution of the agents is, no one will manipulate. The learner can then deploy $f$ for optimal accuracy.

---

> ### Author Response · Authors · 2024-08-07
> **Part 2 of Rebuttal**
>
> Part 2:
>
> **Can truthful information sharing (by the learner) can be enforced or incentivized?:**
>
> Here, we highlight the typical argument in Bayesian Persuasion: if the learner is not truthful, over time, the learner will lose credibility and will not be trusted by the agents anymore. As a result the agents will simply ignore the information coming from the subset release and will base their response according to the prior, which could then become sub-optimal for the learner. We also emphasize that in the applications that we consider in our paper (such as hiring, loans, etc.), while the learner does not need to reveal their model fully, the learner could face legal and ethical challenges if they choose to misrepresent the model for their own favor.
>
>
> **Estimating the feature-label distribution from samples (instead of knowing the distribution) (What are the consequences of the learner not knowing the feature-label distribution $D$, and being required to estimate it from data?):**
>
> Given a data set sampled $i.i.d.$ from the distribution, the learner can find the optimal subset $H$ with respect to the data set and then appeal to standard generalization guarantees to argue that the same subset is (approximately) optimal with respect to the distribution, provided that the sample size is large enough. We note that while this gives only a sketch of the extension from knowing $D$ to having only a sample from $D$, the exact characterization of the sample complexity requires careful analysis and considerations.
>
> **What is the complexity of H, i.e., cardinality of H? (Can the analysis framework be used to estimate the typical cardinality/"complexity" of the optimal hypothesis sets $H$ reported by the learner? Is there intuition for practical scenarios where a non-trivial $H$ is expected to be "simple to describe" or particularly complex in some sense?):**
>
> Our analysis and examples show that the optimal partial information $H$ could in fact be any subset of the hypothesis class. It is not clear to us whether the size of the optimal $H$ can be estimated. Example 2.1 in the paper shows a set of examples in which a subset $H$ is “simple to describe” (like releasing the “relevant features” of the deployed classifier), but we note that such examples for partial information release are not necessarily optimal for the learner.
>
> [1] Kremer, Ilan, Yishay Mansour, and Motty Perry. “Implementing the ‘Wisdom of the Crowd.’” Journal of Political Economy 122, no. 5 (2014): 988–1012. https://doi.org/10.1086/676597.
>
> [2] Yahav Bechavod, Chara Podimata, Steven Wu, Juba Ziani.  “Information Discrepancy in Strategic Learning”. Proceedings of the 39th International Conference on Machine Learning, PMLR 162:1691-1715, 2022.
>
> [3] Kamenica, Emir, and Matthew Gentzkow. "Bayesian persuasion." American Economic Review 101, no. 6 (2011): 2590-2615.
>
> [4] Mark Braverman and Sumegha Garg. 2020. The Role of Randomness and Noise in Strategic Classification. In Foundations of Responsible Computing (FORC) (LIPIcs, Vol. 156).
>
> [5] Dasgupta, S. (2004). Analysis of a greedy active learning strategy. Advances in neural information processing systems, 17.

---

> > ### Comment · Reviewer_oMrT · 2024-08-14
> >
> > Thank you for the detailed response! The answers are very helpful, and I maintain my current rating.

---

### Official Review · Reviewer_F6uC · 2024-07-17

**Soundness:** 4
**Presentation:** 4
**Contribution:** 4
**Rating:** 8
**Confidence:** 3

**Summary:**

The paper introduces a novel framework for strategic classification using a Bayesian setting for agents' beliefs about the classifiers. This framework departs from the traditional assumption that agents have complete knowledge of the deployed classifier and instead assumes that agents have a prior distribution over the possible classifiers. The main components of the model are a population of agents and a learner. Each agent is represented by a pair $(x, y)$, where $x \in X$ is a feature vector and $y \in \{0, 1\}$ is a binary label. An agent with $y = 0$ is called a “negative,” and an agent with $y = 1$ is called a “positive.” There exists a mapping $f: X \to \{0, 1\}$ that governs the relationship between $x$ and $y$, meaning $y = f(x)$ for every agent $(x, y)$.

The model includes the following key components:

1. **Agent Manipulations**: Each agent can manipulate their features, paying a cost function $c: X \times X \to [0, \infty)$, incurred when changing their features from $x$ to $x'$.

2. **Partial Knowledge of Agents**: Agents have a common prior distribution $\pi$ over the hypothesis class $H \subseteq \{0, 1\}^X$, representing their belief about the deployed classifier $h$. Formally, for every $h' \in H$, $\pi(h')$ is the probability that the learner is deploying $h'$ from the agents’ perspective.

3. **Partial Information Release by the Learner**: The learner can release partial information about the classifier to influence agents' beliefs. This is modeled by releasing a subset $H' \subseteq H$ such that $h \in H'$.

4. **Strategic Game with Partial Information Release**: After the partial information is released, each agent computes their posterior belief and then moves to a new point that maximizes their utility.

The paper provides the following theoretical contributions:

1. **Oracle-Efficient Algorithms**: For low-dimensional linear classifiers, the authors present an oracle-efficient algorithm for computing the best response of agents. When $X = \mathbb{R}^d$ and $H$ contains only linear classifiers of the form $h(x) = 1[w^\top x + b \geq 0]$, the best response of the agents can be computed with $O(n^d)$ oracle calls, where $n$ is the number of linear classifiers and $d \ll n$.

2. **Theoretical Analysis of Learner’s Optimization**: The learner’s optimization problem is analyzed to maximize expected classification accuracy under the agents' strategic manipulations. The analysis considers various information release strategies and their impact on agents’ beliefs and manipulations.

3. **Examples and Scenarios**: The paper provides practical examples, such as job seekers using platforms like Glassdoor to form beliefs about hiring algorithms, which ground the theoretical concepts in real-world scenarios.

#### Mathematical Formulation

1. **Agents and Learner**:
    - Agents: $(x, y)$ where $x \in X$, $y \in \{0, 1\}$.
    - Mapping: $f: X \to \{0, 1\}$ such that $y = f(x)$.
    - Hypothesis class: $H \subseteq \{0, 1\}^X$, with the learner using a fixed classifier $h \in H$.

2. **Cost Function**:
    $$
    c: X \times X \to [0, \infty) \quad \text{where} \quad c(x, x') \text{ denotes the cost of changing } x \text{ to } x'.
    $$

3. **Prior Distribution**:
    $$
    \pi: H \to [0, 1] .
    $$


5. **Oracle-Efficient Algorithm**:
    - For linear classifiers $h(x) = 1[w^\top x + b \geq 0]$, with $d$ much smaller than $n$ , the algorithm partitions $X$ given by $n$ linear classifiers into $O(n^d)$ elements and computes the best response within each partition element.
    - The best response algorithm runs in $O(n^{d+1})$ time, making $O(n^d)$ oracle calls.

In summary, this paper presents a comprehensive framework for Bayesian strategic classification, providing both theoretical insights and practical algorithms. The introduction of a probabilistic aspect to agents’ beliefs and the exploration of partial information release by the learner are key contributions that enhance the understanding and modeling of strategic behavior in classification settings.

**Strengths:**

- Bayesian modeling: shifting to a Bayesian framework where agents have a distributional prior on the classifier is a novel and realistic approach, providing a more flexible model than the standard full knowledge assumption.
- Principled algorithms: the paper introduces oracle-efficient algorithms for low-dimensional linear classifiers and sub-modular cost functions, which are valuable contributions to strategic classification.
- Mathematical and algorithmic analysis: the comprehensive analysis of the learner’s optimization problem, including conditions for the optimality of partial information release, adds depth to the study.
- Real-world examples: practical examples, such as job seekers using Glassdoor, help ground the theoretical concepts in real-world scenarios.

**Weaknesses:**

- Realizability Assumption: The assumption that agents’ priors are realizable (i.e., the classifier deployed by the learner is in the support of the agents' prior) may not hold in many practical situations.
- Fixed Classifier Assumption: The learner’s commitment to a fixed classifier limits the model's applicability in dynamic environments where classifiers are frequently updated.
- Empirical analysis: the paper could be enriched with an empirical study of the methods discussed theoretically.

**Questions:**

- The focus on maximizing accuracy is essential, but how do the proposed methods perform with other utility functions, such as fairness, robustness, or cost-efficiency?
- How do the proposed algorithms and strategies perform in empirical settings? Are there practical examples or case studies that demonstrate their applicability and effectiveness?
- How can the model be extended to dynamic environments where classifiers are updated over time?
- Is there specific conditions where partial information release outperforms full information release? Are there scenarios where partial information could be detrimental?
- How does the Oracle complexity algorithm scale with higher dimensions or larger hypothesis classes when $d \approx n$? Are there potential computational bottlenecks?

**Limitations:**

The authors discuss the limitations of the methods adequately.

---

> ### Author Rebuttal · Authors · 2024-08-07
>
> We thank the reviewer for suggesting interesting questions that can be natural future directions of this work such as other utility functions and dynamic environments.
>
> - Regarding the performance of partial information release versus full information release, while we have examples where one outperforms the other, the exact characterization of instances in which one outperforms the other is not known. This is indeed an interesting question.
>
> - For any $n$ and $d$, the best response algorithm for linear classifiers (Algorithm 2) has oracle complexity of $\sum_{j=0}^d {n \choose j}$ which is $O(n^d)$ for $d \ll n$ and $O(2^n)$ for $d \approx n$. This is why we need that the dimension is relatively small compared to $n$.

---

### Decision · Program_Chairs · 2024-09-25

**Decision:**

Accept (poster)

**Comment:**

After the author response and discussion, all the reviewers felt positively about the paper (to varying degrees) with some quite strongly enthusiastic about the paper.  Overall the new model introduced seems to make a nice contribution to this growing and important literature.